# Exploration of Food Attitudes and Management of Eating Behavior from a Psycho-Nutritional Perspective

**DOI:** 10.3390/healthcare12191934

**Published:** 2024-09-27

**Authors:** Anca Mihaela Dicu, Lavinia Denisia Cuc, Dana Rad, Andreea Ioana Rusu, Andrea Feher, Florin Lucian Isac, Daniel Manate, Annamaria Pallag, Florentina Simona Barbu

**Affiliations:** 1Faculty of Food Engineering, Tourism and Environmental Protection, Aurel Vlaicu University of Arad, 310025 Arad, Romania; anca1474@yahoo.com; 2Centre for Economic Research and Consultancy, Faculty of Economics, Aurel Vlaicu University of Arad, 310025 Arad, Romania; laviniacuc@yahoo.com (L.D.C.); florin.isac@uav.ro (F.L.I.); daniel.manate@uav.ro (D.M.); florentina.barbu@uav.ro (F.S.B.); 3Center of Research Development and Innovation in Psychology, Faculty of Educational Sciences Psychology and Social Sciences, Aurel Vlaicu University of Arad, 310025 Arad, Romania; dana@xhouse.ro; 4Faculty of Pharmacy, “Vasile Goldiș” Western University of Arad, 310045 Arad, Romania; 5Department of Economy and Firm Financing, University of Life Sciences ”King Mihai I” from Timisoara, 300645 Timisoara, Romania; 6Research Center for Sustainable Rural Development of Romania, Romanian Academy—Branch of Timisoara, 300223 Timisoara, Romania; 7Department of Pharmacy, Faculty of Medicine and Pharmacy, University of Oradea, 410028 Oradea, Romania; annamariapallag@gmail.com

**Keywords:** food preoccupation, eating behavior, bulimia, anorexia, psycho-nutrition, emotional food craving, healthy lifestyle

## Abstract

Background/Objectives: This study investigates the relationship between food attitudes and the management of eating behavior from a psycho-nutritional perspective, with a focus on the Health Belief Model (HBM). The objective was to explore how emotional food cravings influence different aspects of eating behavior and dietary attitudes, and to identify indirect pathways through which these cravings affect attitudes toward dieting behaviors. Methods: Data were collected from 659 participants using validated scales that assessed dieting tendencies, bulimia and food preoccupation, culinary behaviors, food preoccupation, expectations of positive outcomes, and emotional food cravings. Descriptive statistics, Pearson’s correlations, and network analysis were employed to uncover significant associations among these variables. A sequential mediation analysis was conducted using SPSS PROCESS Macro Model 6 to identify indirect pathways. Results: The analysis revealed significant associations among the variables. Emotional food craving had a negative indirect effect on dieting attitudes through bulimia tendencies (effect size: −0.523) and a positive indirect effect through food preoccupation (effect size: 0.1006). These results highlight the complex interplay between emotional food cravings, bulimia tendencies, and food preoccupation in shaping dieting attitudes. Conclusions: The findings emphasize the complex dynamic between emotional food cravings, bulimia tendencies, and food preoccupation, and their collective impact on dieting attitudes. This study provides insights into potential intervention strategies aimed at improving eating habits by addressing emotional food cravings and their indirect effects on dietary behavior.

## 1. Introduction

Eating behaviors are influenced by a complex dynamic of psychological and nutritional factors [1,2]. Emotional food cravings, in particular, are a powerful driver of eating behavior. These cravings typically manifest as an intense desire or urge to consume specific foods, often triggered by emotional states such as stress, anxiety, boredom, or even happiness [1]. Emotional food cravings are not driven by physiological hunger but rather by the need to cope with or soothe emotions, leading individuals to turn to food as a source of comfort. The foods craved in these situations tend to be high in sugar, fat, or carbohydrates, which are known for their ability to temporarily elevate mood or provide feelings of pleasure. However, these temporary reliefs often come at a cost, as they can promote unhealthy eating patterns, such as overeating or the development of restrictive dieting practices. These maladaptive responses to emotional food cravings can complicate individuals’ relationships with food, influencing their attitudes toward dieting and body image. 

Previous research has primarily focused on dietary habits, dieting tendencies, and body image concerns, but there is a gap in exploring emotional food cravings within the Health Belief Model (HBM) framework. Akey, Rintamaki, and Kane [3] applied the HBM to study the barriers to seeking social support among individuals with eating disorders, showing how perceived barriers and susceptibility shape eating behaviors. Grodner [4] used the HBM to design bulimia prevention programs, underscoring the importance of perceived severity and benefits in motivating behavior changes. Hepworth and Paxton [5] demonstrated that help-seeking behaviors for bulimia and binge eating are influenced by individuals’ perceptions of susceptibility and severity.

The HBM serves as the theoretical foundation for this study. According to the HBM, individuals’ health-related actions are influenced by their perceptions of susceptibility to health issues, the severity of those issues, the benefits of taking preventive actions, and the barriers to taking such actions [6,7]. This model has been applied to various health behaviors but remains underexplored in the context of emotional eating and food cravings. The HBM framework is relevant for understanding how emotional food cravings shape dietary attitudes by influencing individuals’ perceptions of their susceptibility to weight-related issues, beliefs about the severity of these issues, and the perceived benefits of healthier behaviors.

This study addresses the gap by integrating emotional food cravings into the HBM framework to examine how these cravings influence dietary attitudes and behaviors. Using validated scales to assess key dimensions related to eating behavior—including dieting tendencies, bulimia and food preoccupation, culinary behaviors, expectations of positive outcomes, and emotional food cravings—this research uncovers the indirect pathways through which emotional food cravings affect dietary attitudes. By applying advanced statistical techniques, such as network analysis and sequential mediation analysis, the study provides a comprehensive understanding of the psycho-nutritional factors influencing eating behavior.

The present study tests the hypothesis stating that emotional food cravings influence dieting behaviors through a sequential mediation pathway involving bulimia and food preoccupation, culinary behaviors, food preoccupation, and expectations of positive outcomes, with these variables collectively shaping the impact of emotional cravings on dietary attitudes and practices within the HBM framework. The statistical methods employed in this study are designed to rigorously test the mediation hypothesis within the framework of the HBM. Specifically, the study aims to examine the sequential pathways through which emotional food cravings influence dieting behaviors via mediators such as bulimia and food preoccupation, culinary behaviors, and expectations of positive outcomes. The use of mediation analysis allows for the identification of indirect effects, providing a detailed understanding of how these psychological and behavioral factors interact to shape dietary attitudes and practices. By aligning these methods with the HBM, the study provides a robust theoretical and empirical basis for testing these complex relationships.

## 2. Literature Review

Emotional food cravings are a central factor in understanding how individuals regulate their eating behaviors. These cravings, which often arise in response to emotional triggers such as stress or anxiety, can lead to maladaptive eating patterns, including overeating or restrictive dieting. Understanding emotional food cravings is crucial for examining their broader influence on eating attitudes, dietary choices, and overall nutritional health [8,9,10,11,12,13]. This review synthesizes current research on emotional food cravings and their impact on eating behaviors, with particular attention to how they interact with psychological and nutritional factors to shape dietary attitudes. Additionally, the review explores related topics such as healthy lifestyles, psycho-nutrition, appetite regulation, and the mediation pathways through which emotional food cravings influence dieting behaviors.

### 2.1. Healthy Lifestyle

A healthy lifestyle encompasses behaviors like regular physical activity, balanced nutrition, and avoiding harmful habits. These behaviors are critical for reducing morbidity and mortality, particularly in individuals with obesity. Matheson, King, and Everett [14] demonstrated that adopting healthy behaviors significantly reduces mortality risk, underscoring the importance of lifestyle modifications for extending lifespan and mitigating health risks. This connects to the present study’s focus on how emotional food cravings disrupt otherwise healthy lifestyle choices, potentially exacerbating unhealthy dieting behaviors.

Education also plays a critical role in fostering healthy behaviors. Park and Kang [15] showed that higher levels of education correlate with healthier lifestyle choices, highlighting how informed decision-making influences attitudes toward dieting. Abdullayeva and Ismailova [16] similarly found that health education enhances public awareness and participation in health-promoting activities. These findings emphasize the importance of cognitive and educational factors in shaping food-related attitudes, which align with the current study’s exploration of how emotional cravings mediate the relationship between food attitudes and dieting behaviors.

In terms of broader lifestyle impacts, Ford and collaborators [17] and Cîrnaţu et al. [18] emphasize that healthy behaviors, such as regular exercise and balanced diets, lower all-cause mortality and foster longevity. These findings are relevant for understanding how emotional cravings can impede or support these health outcomes, particularly as emotional food cravings may undermine individuals’ ability to maintain healthy lifestyle choices.

While Cockerham’s health lifestyle theory [19] explains the interaction of personal and social factors on health behaviors, the current study uses the HBM to focus specifically on how personal beliefs about susceptibility to health risks influence eating attitudes and behaviors. The HBM framework is more applicable in examining how emotional cravings, acting as a mediator, affect dietary behavior by modulating perceptions of health risks and benefits.

### 2.2. Psycho-Nutrition

Psycho-nutrition explores the intersection between psychological factors and dietary behaviors to improve mental health through nutrition [20,21]. Greenbaum [22] found that dietary habits profoundly affect psychological well-being. This field is critical for understanding how emotional food cravings, as a psychological phenomenon, influence eating behavior. Emotional cravings, often driven by stress or negative emotions, can lead to unhealthy eating patterns, which is a central focus of this study.

Achour and colleagues [23] emphasized the importance of integrating psychological and nutritional techniques in psychiatric care, highlighting how addressing both mental health and dietary behaviors improves patient outcomes. This intersection is relevant for understanding the mediation role of emotional food cravings in the relationship between psychological states and dieting behaviors.

Workshops combining dietary education with psychological support have been shown to significantly improve emotional and physical health, especially in individuals dealing with obesity and emotional eating [24]. This finding is directly relevant to the current study’s exploration of how emotional cravings impact dieting behaviors. Psycho-nutritional interventions, such as those described by Schlienger [25], have been successful in treating eating disorders by addressing the fear of weight gain and distorted body image perceptions. This reflects the study’s aim of understanding how emotional food cravings may contribute to bulimia tendencies and food preoccupation.

### 2.3. Influences of Eating Behavior

Eating behavior is shaped by a range of factors, including regulatory systems, addiction-like tendencies, and emotional and cultural influences [26,27,28,29,30,31,32,33,34]. Teixeira et al. [26] validated the Regulation of Eating Behavior Scale (REBSp), which measures both internal motivations and external influences on eating. This is directly relevant to the study’s exploration of how emotional cravings interact with bulimia tendencies, food preoccupations, and dieting attitudes.

Addictive eating behaviors, as described by Legendre and Bégin [27], highlight how some individuals exhibit food addiction patterns that disrupt healthy eating behaviors. These addiction-like behaviors are essential to the current study, as emotional food cravings may act as a gateway to compulsive eating. Pinho et al. [29] explored barriers to healthy eating, such as cost and accessibility, which also intersect with emotional drivers of unhealthy eating. Cultural factors, as discussed by Riou et al. [30], further shape eating patterns and attitudes, which may affect how emotional cravings are expressed in different populations.

### 2.4. Eating Attitudes: Bulimia and Anorexia

Eating attitudes, particularly those involving bulimia and anorexia, provide critical insights into how emotional food cravings might escalate into severe disordered eating patterns. The Eating Attitudes Test (EAT), reviewed by Garfinkel and Newman [35], is essential in detecting early signs of problematic eating behaviors. These tools are important in the current study, as they help understand how emotional food cravings influence dieting behaviors by intensifying preoccupation with food and body image.

Goldschmidt et al. [36] found that weight-related preoccupations often lead to unhealthy eating attitudes among children. Early identification and intervention, as described by Maloney et al. [37], can prevent more severe eating disorders later in life. These insights align with the study’s focus on understanding how early emotional food cravings may lead to problematic eating behaviors such as bulimia.

### 2.5. Appetite

Appetite is shaped by a complex interaction of genetic, environmental, and behavioral factors [38,39,40,41,42]. Brunner et al. [43] found that genetic predispositions influence hunger disinhibition, which contributes to obesity. These findings highlight the need to consider how emotional food cravings can disrupt normal appetite regulation and lead to overeating. Appetite is also affected by early life experiences, as shown by Jansen et al. [44], which underscores the importance of early interventions in shaping healthy food attitudes and dieting behaviors later in life [45].

Dalton et al. [46] emphasized that individuals with obesity often exhibit behavioral phenotypes, such as heightened sensitivity to food rewards, which complicates appetite regulation. This has direct implications for the study’s examination of how emotional cravings affect appetite control and dieting behavior. While the Behavioral Susceptibility Theory [47] explains the genetic and environmental interactions in hunger management, the current study focuses more on the emotional and psychological influences on appetite and dieting behaviors.

### 2.6. Emotional Food Craving

Emotional food cravings involve a complex interplay between emotional states and eating behaviors, often leading to maladaptive eating patterns [48,49,50]. Emotional states such as stress, boredom, or anxiety have been shown to trigger uncontrolled eating [51], which is central to the current study’s examination of how emotional cravings affect dieting behaviors. Waters, Hill, and Waller [52] found that emotional discomfort often triggers binge-eating episodes, providing evidence that emotional drivers, rather than physiological hunger, can dominate eating behaviors.

Janse Van Vuuren et al. [53] demonstrated that emotional cravings negatively affect post-surgical weight management outcomes. This aligns with the study’s aim of exploring how emotional food cravings undermine long-term dieting success. Furthermore, psychological interventions such as cognitive behavioral therapy (CBT) and mindfulness have been shown to mitigate the effects of emotional cravings on eating behaviors [54,55,56,57,58]. These findings are critical in understanding the psychological factors that mediate the relationship between emotional food cravings and dieting behaviors.

### 2.7. Regulation of Eating Behavior

The regulation of eating behavior involves multiple psychological, cognitive, and physiological processes that are crucial for managing food intake and maintaining a balanced diet. Johnson, Pratt, and Wardle [59] explored how dietary restraint and self-regulation impact eating behaviors. They found that individuals who struggle with self-control are more likely to experience difficulties in regulating their eating patterns, particularly in response to emotional triggers. This finding is relevant to the current study, as emotional food cravings may disrupt self-regulation, leading to overeating or unhealthy dieting behaviors.

Dohle, Diel, and Hofmann [60] examined the role of executive functions, such as inhibition, planning, and decision-making, in the regulation of eating behaviors. These cognitive processes are essential for effective self-regulation, allowing individuals to control their food intake even when faced with emotional or environmental cues that might trigger cravings. This research aligns with the current study’s focus on how emotional food cravings undermine cognitive control, leading to maladaptive eating patterns.

Motivational aspects of eating behavior are also critical in determining how individuals regulate their food intake [61]. Pelletier et al. [62] explored how intrinsic and extrinsic motivations influence long-term dietary changes and psychological well-being. Their research suggests that understanding the motivational components behind eating behaviors can inform the development of interventions aimed at promoting healthier eating patterns. This aligns with the current study’s exploration of how emotional food cravings, which may be driven by both intrinsic and extrinsic motivations, disrupt the regulation of eating behaviors.

Environmental cues also play a role in influencing eating behaviors. Papies and Hamstra [63] found that goal priming, where environmental signals activate specific eating-related goals, can enhance self-regulatory control over food intake. This highlights the potential for using environmental and contextual interventions to support better eating regulation, especially in individuals who experience strong emotional food cravings. By understanding how environmental factors interact with cognitive and motivational processes, this study can offer a more comprehensive perspective on how emotional food cravings disrupt eating behavior regulation.

Finally, external stressors, such as those experienced during the COVID-19 pandemic, have been shown to significantly influence self-regulation in eating behaviors. Guzek, Skolmowska, and Głąbska [64] explored how stress and uncertainty during the pandemic led to changes in dietary habits, with many individuals reporting an increase in emotional eating. This underscores the importance of addressing emotional factors when developing interventions to improve eating behavior regulation, particularly in times of heightened stress.

The gender differences in eating behavior regulation, as examined by Rolls, Fedoroff, and Guthrie [65], also have implications for the current study. Their research identified physiological and cultural factors that contribute to differences in how men and women regulate their eating behaviors. These differences must be considered when interpreting the results of this study, as emotional food cravings may manifest differently across genders, influencing the overall regulation of eating behaviors.

In summary, the regulation of eating behaviors is shaped by a combination of psychological, cognitive, and physiological processes, with emotional food cravings playing a disruptive role. By integrating insights from studies on self-regulation, cognitive processes, hormonal regulation, and motivational factors, the current study aims to explore how emotional food cravings interfere with the regulation of eating behaviors, ultimately influencing dieting attitudes and outcomes. This comprehensive understanding is critical for developing targeted interventions that can mitigate the impact of emotional cravings on eating behavior.

## 3. Methodology

This study is grounded in the HBM, a psychological framework widely used to understand health-related behaviors [66]. The HBM posits that individuals’ health behaviors are influenced by their perceptions of susceptibility to health problems, the perceived severity of these problems, the perceived benefits of taking preventive actions, and the perceived barriers to taking these actions. This model also considers cues to action and self-efficacy as critical factors in motivating behavior change.

These mediators are conceptualized within the HBM as follows:Perceived susceptibility and severity: Bulimia and food preoccupation, as measured by the Eating Attitude Test, reflect individuals’ perceptions of their susceptibility to and the severity of disordered eating behaviors. This aligns with the HBM’s focus on how perceived health threats influence behavior.Perceived benefits and barriers: Emotional food cravings and the expectation of positive outcomes, assessed using the General Food Cravings Questionnaire—Trait, correspond to the perceived benefits and barriers in the HBM. These factors determine the likelihood of individuals engaging in dieting behaviors as a means of achieving desired health outcomes or avoiding negative ones.Cues to action: Culinary behaviors, also measured by the Eating Attitude Test, can be seen as cues that prompt individuals to engage in specific health-related actions, such as modifying their diet or eating patterns.Self-efficacy: The belief in one’s ability to successfully alter eating behaviors, although not directly measured, is implicit in the respondents’ attitudes towards food and dieting, as captured in the scales used.

By structuring the analysis within the HBM framework [67], this study aims to provide a comprehensive understanding of how psychological factors and health beliefs shape dietary behaviors. This approach not only enhances the interpretation of findings but also situates them within a broader context of health behavior research.

The objective of this paper is to investigate the relationships between food attitudes and the management of eating behavior from a psycho-nutritional perspective. Specifically, the study aims to explore how emotional food cravings influence attitudes toward dieting behaviors through sequential mediation pathways involving bulimia tendencies, culinary behaviors, food preoccupations, and expectations of positive outcomes.

This analytical approach allows for a detailed examination of how emotional food craving (X) sequentially mediates pathways through multiple intermediate variables: bulimia and food preoccupation (M1), culinary behaviors (M2), food preoccupation (M3), and expect positive results (M4). The ultimate goal is to identify the sequential chain of effects that these variables exert on dieting behaviors (Y).

### 3.1. Instruments

The selection of the Eating Attitude Test (EAT-26) and the General Food Cravings Questionnaire—Trait (G-FCQ-T) as primary measurement tools in this study is well-aligned with the theoretical underpinnings of the HBM. The EAT-26, which assesses dieting behaviors, bulimia, and food preoccupation, effectively captures individuals’ perceived susceptibility and severity regarding disordered eating patterns. Meanwhile, the G-FCQ-T, which measures emotional food cravings and expectations of positive outcomes, provides insights into perceived benefits, barriers, and self-efficacy related to food consumption. These scales collectively offer a comprehensive framework to evaluate the psychological and behavioral factors that the HBM posits are critical in shaping health-related behaviors. Thus, their use in this study facilitates a detailed examination of the beliefs and attitudes that influence eating behaviors, aligning with the core constructs of the HBM and supporting the study’s objectives.

The EAT-26, developed by Garner and collaborators [40] in 1982, is a shortened version derived from the original EAT-40 through factor analysis. EAT is a comprehensive instrument designed to evaluate attitudes and behaviors associated with eating and body image. It is widely employed in both research and clinical settings to identify symptoms indicative of eating disorders. The EAT is structured into two primary sections, each addressing different dimensions of eating attitudes and behaviors. The first section comprises a variety of statements that participants respond to, indicating the frequency with which they experience specific thoughts and behaviors related to eating and body image. These responses are rated on a 1 to 6 Likert scale ranging from “Always” to “Never”, allowing for the capture of the intensity and regularity of these experiences. This section is essential in identifying general tendencies and attitudes towards food and body weight, offering insights into the psychological aspects of eating behaviors. The second section focuses on specific behaviors that participants have engaged in over the past six months, such as binge eating, self-induced vomiting, and the use of laxatives or diuretics for weight control. This part aims to gather concrete data on the frequency of behaviors typically associated with eating disorders, providing a more direct assessment of the participant’s actions rather than merely their attitudes.

The EAT is divided into several subscales, each targeting distinct aspects of eating attitudes and behaviors. The dieting subscale examines behaviors and thoughts related to dieting and weight control, with items such as “I am terrified about being overweight” and “I am aware of the calorie content of foods I eat”. These items help gauge the extent of dieting behaviors and the preoccupation with body image, which are crucial for understanding how preoccupations about weight and dieting influence eating habits.

Another subscale is the Bulimia and food preoccupation subscale, which assesses tendencies towards bulimic behaviors and an obsession with food. Example items from this subscale include “I have gone on eating binges where I feel that I may not be able to stop” and “I find myself preoccupied with food”. These items are essential for identifying behaviors and thoughts related to binge eating and the psychological aspects of food obsession.

For the purposes of this study, only the Dieting subscale and the Bulimia and food preoccupation subscale were utilized. These subscales were specifically selected to focus on dieting behaviors and food preoccupation, aligning with the study’s objectives of exploring the psychological and behavioral dimensions of dieting and emotional food cravings.

The EAT demonstrated good internal consistency, with a Cronbach’s alpha coefficient of 0.858, indicating strong reliability. When based on standardized items, Cronbach’s alpha remains high at 0.861. The scale has a mean score of 110.55, a variance of 336.99, and a standard deviation of 18.36, reflecting moderate variability in responses among participants. Hotelling’s T-Squared test was conducted to assess the overall multivariate significance of the scale, yielding a significant result (T^2^ = 4364.48, F(25, 634) = 168.21, *p* < 0.001), indicating that the items collectively measure distinct aspects of eating attitudes reliably.

The General Food Cravings Questionnaire—Trait (G-FCQ-T), adapted from the original multidimensional Trait and State Food Cravings Questionnaires (FCQ-T and FCQ-S), was developed by Cepeda-Benito and collaborators [67] in 2000 and modified by Nijs, Franken and Muris [57] in 2007. This 21-item questionnaire focuses on general food cravings across various contexts rather than specific food items. This instrument is widely used in both clinical and research contexts to understand the complexities of food cravings, particularly in relation to emotional and cognitive factors. Participants rate their responses on a 6-point Likert scale ranging from “never applies to me” to “always applies to me”.

The first factor, Food preoccupation, explores the extent to which individuals are preoccupied with thoughts about food. It includes items such as “I think about food all the time”, “I feel that I cannot stop thinking about food, no matter how hard I try”, and “I am preoccupied with food”. These items help to gauge the cognitive intrusion of food-related thoughts into an individual’s daily life, providing insights into the mental patterns associated with food cravings.

The second factor, Culinary behaviors, assesses the individual’s perceived loss of control over eating behaviors, particularly in response to cravings. Example items include “When I start eating, it is hard to stop”, “If I eat what I crave, I often lose control and eat too much”, and “When I crave something, I know I won’t be able to stop once I start eating”. These items are critical for understanding the compulsive aspects of eating behavior that often accompany intense cravings.

The third factor, Expectation of a positive outcome, measures the positive reinforcement and emotional satisfaction derived from eating, particularly in the context of satisfying cravings. Items such as “When I eat what I crave, I feel better”, “When I eat what I crave, I feel great”, and “I feel less anxious after I eat” capture the emotional rewards associated with food consumption, highlighting the affective motivations behind food cravings.

The final factor, Emotional food craving, examines the role of emotions in triggering cravings. Items include “I crave certain foods when I am upset”, “My emotions often make me want to eat”, and “When I am stressed, I crave food”. This factor is essential for understanding how emotional states can influence eating behaviors, often leading to emotional eating.

The G-FCQ-T demonstrates excellent internal consistency with a Cronbach’s alpha coefficient of 0.944, indicating strong reliability. When based on standardized items, Cronbach’s alpha remains high at 0.945. The questionnaire has a mean score of 56.31, a variance of 540.45, and a standard deviation of 23.25 across its 21 items, suggesting moderate variability in responses among participants. Hotelling’s T-Squared test was conducted to assess the overall multivariate significance of the G-FCQ-T, yielding a significant result (T^2^ = 833.89, F (20, 639) = 40.49, *p* < 0.001), indicating that the questionnaire reliably captures distinct aspects of food cravings.

### 3.2. Participants

Participants were recruited through convenience sampling via a Google Form questionnaire administered from May to June 2024. Regarding ethical considerations, this study adhered to the ethical guidelines outlined by the CCDIP of Aurel Vlaicu University of Arad. Ethical approval for the study was obtained from the CCDIP, which ensures that the research adhered to established ethical standards and protocols. Participants provided informed consent before completing the questionnaire, and all data were collected and handled in accordance with ethical research practices to protect participant confidentiality and privacy.

A total of 659 individuals completed the questionnaire, representing a diverse array of demographics and characteristics from a subclinical population in Western Romania. The survey was distributed through various social media groups to ensure broad outreach.

The sample for this study predominantly consisted of females (81.2%, n = 535 vs. males: 18.8%, n = 124), which reflects a gender imbalance. This distribution may be attributed to the tendency for women to be more responsive to open questionnaires, and women often exhibit higher engagement with such surveys compared to men. Despite this imbalance, the sample was deemed representative for the purpose of this study, as the primary focus was not on gender differences but rather on the overall relationships between emotional food cravings and dieting behaviors within the subclinical population of Western Romania. The data collection process, which involved distributing the questionnaire via various social media groups, was designed to reach a broad audience, thus enhancing the generalizability of the findings within the context of the study’s specific aims.

Participants were from primarily urban (69.3%, n = 457) and rural (30.7%, n = 202) areas, indicating a mix of residential backgrounds. In terms of educational attainment, 51.6% (n = 340) had completed secondary education, 26.6% (n = 175) had completed higher education, 19.9% (n = 131) held undergraduate degrees, and 2.0% (n = 13) held postgraduate degrees.

Table 1 presents descriptive statistics for age, height, weight, and income among participants, along with gender, residential area, and educational level. Age ranged from 16 to 66 years (mean [M] = 31.16, SD = 11.967). Heights ranged from 152.00 cm to 190.00 cm (M = 171.22 cm, SD = 38.95616). Weights ranged from 40.00 kg to 163.00 kg (M = 69.14 kg, SD = 17.02434). Income, measured in Romanian Leu (RON), had a mean of 3727.46 RON (SD = 2500.628), with values ranging from 0 to 15,000 RON. In order to contextualize the economic background of participants, this study categorized their monthly incomes into three groups: low income (0–2000 RON), medium income (2001–5000 RON), and high income (above 5000 RON). The distribution of participants across these income categories was as follows: 31% fell into the low-income category, 48% into the medium-income category, and 21% into the high-income category.

### 3.3. Data Analysis

The data collected for this study were subjected to a comprehensive analysis encompassing descriptive statistics, correlation analysis, and network analysis, followed by sequential mediation analysis using SPSS V.26 PROCESS Macro Model 6 [68].

Initially, descriptive statistics were employed to summarize the demographic characteristics of the participants and key psycho-nutritional variables: dieting behaviors, bulimia and food preoccupation, culinary behaviors, food preoccupation, expectations of positive results, and emotional food cravings. Measures such as mean, standard deviation, minimum, and maximum were computed to provide an overview of the data distribution.

Pearson’s correlations were calculated to explore the relationships among emotional food craving, dieting tendencies, bulimia and food preoccupation, culinary behaviors, food preoccupations, and expectations of positive outcomes. This analysis provided insights into the strength and direction of associations among these variables.

A network analysis was conducted to visualize and quantify the relationships between variables using the igraph package in R. The network diagram depicted nodes representing variables and edges indicating pairwise correlations. Centrality measures such as betweenness, closeness, strength, and expected influence were computed to identify key variables that play pivotal roles within the network structure.

To elucidate the sequential mediation pathways linking emotional food craving to attitudes toward dieting behaviors, SPSS PROCESS Macro Model 6 was utilized. This analysis enabled the estimation of total, direct, and indirect effects, exploring how emotional food craving influences dieting attitudes through mediators including bulimia tendencies, culinary behaviors, food preoccupations, and positive outcome expectations. Bootstrap confidence intervals with 5000 resamples were applied to assess the significance of indirect effects, enhancing the reliability of the mediation pathways identified.

By integrating these analytical approaches, this study provides a comprehensive examination of the psycho-nutritional dynamics influencing attitudes toward dieting behaviors among the study participants.

## 4. Results

The statistical analysis of the subscales revealed notable trends in participants’ eating behaviors. For the dieting subscale, participants showed a relatively high mean score, indicating a moderate level of dieting behavior. Similarly, bulimia and food preoccupation scores suggested moderate levels of bulimic symptoms and food-related preoccupations, while culinary behaviors reflected lower engagement with food preparation activities. Food preoccupation, measured by the General Food Cravings Questionnaire—Trait (G-FCQ-T), indicated moderate concern with food, and participants also showed moderately high expectations of positive outcomes related to food behaviors. Emotional food craving scores pointed to a moderate level of emotional attachment to food. Overall, the findings highlight a range of eating attitudes and behaviors among participants, as detailed in Table 2.

### 4.1. Correlation Analysis

The correlation analysis results using Pearson’s correlation coefficients are presented in Table 3. This research looks at the correlations between the following subscales: dieting, bulimia and food preoccupation, culinary behaviors, food preoccupation, expecting positive results, and emotional food craving. The significance levels are indicated as follows: * *p* < 0.05, ** *p* < 0.01, *** *p* < 0.001.

The correlation analysis, utilizing Pearson’s correlation coefficients, identifies significant associations among various subscales: dieting, bulimia and food preoccupation, culinary behaviors, food preoccupation, expectations of positive results, and emotional food craving. The study reveals a significant positive relationship (r = 0.590, *p* < 0.001) between dieting and bulimia and food preoccupation. This indicates that higher scores in dieting are associated with higher levels of bulimia and food preoccupation, suggesting that individuals who engage in more rigorous dieting may also experience greater levels of these issues.

Conversely, dieting shows a moderately negative correlation with culinary behaviors (r = −0.288, *p* < 0.001) and food preoccupation (r = −0.269, *p* < 0.001). This implies that higher dieting scores are linked to lower engagement in culinary behaviors and less food preoccupation, suggesting that those who diet more strictly may focus less on food-related activities and have fewer food-related concerns.

Additionally, there is a weak negative correlation between dieting and expectations of positive results (r = −0.079, *p* < 0.05), and a moderate negative correlation with emotional food craving (r = −0.310, *p* < 0.001). This indicates that higher dieting scores are slightly to moderately associated with lower expectations of positive outcomes and reduced emotional food craving. In practical terms, this means that those who diet intensively might experience fewer cravings and have lower expectations for positive results from their dietary efforts.

Higher scores in bulimia and food preoccupation are associated with lower scores in culinary behaviors and food preoccupation (r = −0.538, *p* < 0.001). This suggests that individuals with higher bulimia scores may have less engagement in culinary activities and lower levels of food preoccupation, highlighting a potential area where dietary concerns may affect daily eating behaviors.

Culinary behaviors demonstrate a positive correlation with food preoccupation (r = 0.431, *p* < 0.001), indicating that increased engagement in culinary activities is related to higher food preoccupation. This suggests that those who are more involved in food preparation and related activities may have higher levels of concern about food.

Expecting favorable outcomes has a moderate positive correlation with both food preoccupation (r = 0.511, *p* < 0.001) and emotional food craving (r = 0.519, *p* < 0.001). This indicates that individuals who have higher expectations for positive results from their dietary behaviors are also likely to experience greater food preoccupation and emotional food cravings.

Finally, emotional food craving has a substantial positive correlation with food preoccupation (r = 0.592, *p* < 0.001), showing that higher levels of emotional food craving are strongly associated with increased food preoccupation. This means that those who experience more intense emotional cravings for food are also likely to be more preoccupied with food.

### 4.2. Network Analysis

To further explore the relationships among the subscales, a network analysis was conducted to visualize the complex interactions and centrality of each variable within the system. The network comprised six nodes, representing each subscale, and displayed 13 non-zero edges out of a possible 15, reflecting a moderate level of connectivity with a sparsity of 0.133.

In network analysis, betweenness centrality measures a variable’s role in connecting other variables within the network. A higher betweenness centrality indicates that the variable serves as an essential bridge between other variables, facilitating the flow of influence or information. Closeness centrality reflects how close a variable is to all other variables in the network. Variables with higher closeness centrality are more central and accessible, meaning they can influence or be influenced by other variables more readily. Expected influence quantifies the potential impact a variable might have on other variables in the network, combining both its direct influence and its role as a connector. Strength indicates the intensity of connections between a variable and other variables. Positive values suggest stronger connections, while negative values indicate weaker or less direct connections.

Mathematically, higher absolute values in these metrics reflect greater influence or centrality in the network. There is no fixed threshold for interpreting these values because the analysis focuses on relative differences among variables rather than absolute cutoff points. Positive values indicate a higher level of influence or centrality, while negative values suggest lower influence or centrality. Thus, interpretation relies on comparing these values to understand each variable’s role within the network.

The centrality measures for each variable in the network are described as follows:Dieting: This variable exhibited low betweenness (−0.580), indicating that it plays a less central role in connecting other variables within the network. Its low closeness (−0.696) suggests that it is less central in terms of its proximity to other nodes. Both the strength of connections (Strength = −0.020) and expected influence (Expected Influence = 0.123) were minimal, reflecting a limited impact on the overall network.Bulimia and food preoccupation: This variable demonstrated moderate betweenness (0.464) and closeness (0.294), suggesting it has a more central role in connecting other variables and is moderately integrated within the network. The strength of connections (Strength = 0.299) and expected influence (Expected Influence = −1.463) were moderate, indicating a significant but balanced impact on the network.Culinary behaviors: Exhibiting low betweenness (−0.580) and very low closeness (−0.810), this variable plays a minor role in linking other nodes. With the lowest strength of connections (Strength = −1.483) and expected influence (Expected Influence = −0.941), culinary behaviors have minimal impact on the network structure.Food preoccupation: This variable showed high betweenness (1.855) and closeness (1.883), indicating a central role in connecting other variables and a close integration within the network. The high strength of connections (Strength = 1.542) and expected influence (Expected Influence = 0.764) highlight its significant impact on other nodes.Expectation of positive results: The variable demonstrated low betweenness (−0.580) and closeness (−0.307), suggesting less centrality and influence within the network. The strength of connections (Strength = −0.541) and expected influence (Expected Influence = 1.094) were moderate, indicating a moderate impact on the network.Emotional food craving: This variable exhibited low betweenness (−0.580) and closeness (−0.364), indicating less centrality and influence within the network. Nevertheless, with moderate strength of connections (Strength = 0.202) and expected influence (Expected Influence = 0.423), it still exerts a moderate impact on the network.

These measures provide insight into the relative importance and influence of each variable within the network, illustrating how each subscale interacts with others and contributes to the overall system (Table 4).

These centrality measures provide an in-depth understanding of how each variable contributes to the overall structure and dynamics of eating attitudes and behaviors, highlighting their varying roles in the interconnected network (Figure 1).

The network analysis diagram (Figure 1) presents a visual representation of the relationships among the variables studied. Each node represents a variable, and the lines connecting these nodes illustrate the strength and direction of their relationships. Positive correlations are indicated by blue lines, while negative correlations are shown with red lines. The thickness of these lines corresponds to the magnitude of the connections, with thicker lines representing stronger correlations and thinner lines denoting weaker ones. Numbers associated with the lines provide specific details on the strength of these relationships, allowing for accurate interpretation. The use of color and line thickness effectively conveys the nature and intensity of the connections between variables. Blue lines highlight positive relationships where variables increase together, while red lines illustrate negative relationships where an increase in one variable is linked to a decrease in another. This visual approach aids in understanding the centrality and impact of each variable within the network.

### 4.3. Sequential Mediation Analysis

To further analyze these relationships, sequential mediation analysis was performed using the PROCESS macro (Model 6) in SPSS Version 26. This analysis focused on understanding how emotional food craving (X) sequentially mediates pathways through bulimia and food preoccupation (M1), culinary behaviors (M2), food preoccupation (M3), and expecting positive results (M4) to impact dieting (Y).

The total effect (c) of emotional food craving on dieting was significant (c = −0.6087, *p* < 0.001), indicating that higher levels of emotional food craving predict increased dieting behaviors. After accounting for all mediators, the direct effect (c’) of X on Y remained significant (c’ = −0.2330, *p* = 0.0046), suggesting a direct relationship between emotional food craving and dieting (Table 5).

Moreover, sequential mediation revealed significant indirect effects through the proposed mediators as follows.

Indirect Effects: These pathways illustrated how emotional food craving influences dieting through various intermediaries. For instance, the indirect effect through bulimia and food preoccupation (Ind1 = −0.5230) indicated that emotional food craving affects dieting via its impact on bulimic tendencies. Similarly, pathways involving culinary behaviors (Ind2 = −0.0693), food preoccupation (Ind3 = 0.1006), and expecting positive results (Ind4 = 0.0605) further explained the chain of influence from emotional food craving to dieting behavior.

The completely standardized indirect effects provided insights into the relative strength and direction of these pathways, highlighting their role in linking emotional responses to food with specific behaviors related to dieting. This sequential mediation analysis not only confirms the direct impact of emotional food craving on dieting but also identifies the mechanisms through which intermediate variables sequentially mediate this relationship, offering a psycho-nutritional perspective on how emotional factors shape eating attitudes and behaviors.

In Table 6, each mediator’s role was assessed individually within its respective models. Starting with bulimia and food preoccupation, the model revealed a significant intercept of 32.8116 (SE = 0.3496, *p* < 0.001), indicating the expected level of bulimia and food preoccupation when emotional food craving is zero. Emotional food craving exhibited a negative coefficient of −0.3512 (SE = 0.0269, *t* = −13.0589, *p* < 0.001), suggesting that higher emotional food craving was associated with reduced bulimic tendencies. This negative relationship implies that individuals experiencing higher emotional food craving may engage less in bulimic behaviors.

Moving to the culinary behaviors model, the intercept was 15.3776 (SE = 1.6335, *p* < 0.001). Emotional food craving showed a positive coefficient of 0.2349 (SE = 0.0372, *t* = 6.3243, *p* < 0.001), indicating that increased emotional food craving was linked to heightened culinary behaviors. Concurrently, bulimia and food preoccupation exhibited a negative coefficient of −0.2287 (SE = 0.0480, *t* = −4.7617, *p* < 0.001), revealing a relationship where higher bulimic tendencies were associated with reduced engagement in culinary behaviors.

In the food preoccupation model, the intercept was 19.0795 (SE = 1.6912, *p* < 0.001). Emotional food craving demonstrated a positive coefficient of 0.4395 (SE = 0.0372, *t* = 11.8205, *p* < 0.001), indicating that heightened emotional food craving corresponded to increased food preoccupations. Bulimia and food preoccupation exhibited a negative coefficient of −0.4408 (SE = 0.0475, *t* = −9.2870, *p* < 0.001), suggesting that individuals with higher bulimic tendencies reported lower levels of food preoccupations. Additionally, culinary behaviors showed a positive coefficient of 0.2566 (SE = 0.0379, *t* = 6.7643, *p* < 0.001), indicating a positive association with food preoccupations.

Lastly, in the expectations of positive results model, the intercept was 4.5691 (SE = 1.9071, *p* = 0.0169). Emotional food craving had a positive coefficient of 0.3848 (SE = 0.0423, *t* = 9.1054, *p* < 0.001), indicating that increased emotional food craving was associated with higher expectations of positive results. Bulimia and food preoccupation showed a positive coefficient of 0.1297 (SE = 0.0521, *t* = 2.4899, *p* = 0.0130), suggesting a positive relationship where higher bulimic tendencies corresponded to higher expectations of positive outcomes. Conversely, culinary behaviors and food preoccupation exhibited negative coefficients of −0.1085 (SE = 0.0405, *t* = −2.6785, *p* = 0.0076) and 0.3567 (SE = 0.0403, *t* = 8.8464, *p* < 0.001), respectively, indicating that individuals with higher culinary behaviors tended to have lower expectations of positive outcomes, whereas those with greater food concerns were more likely to have higher expectations of positive results. The observed relationship indicates that individuals with higher levels of culinary behaviors tend to have lower expectations of achieving positive outcomes from their dietary practices. Conversely, individuals with higher food concerns, which refer to worries and preoccupations with food, tend to have higher expectations of positive results from their dietary behaviors. This suggests that those more engaged in food-related activities may be less optimistic about the effectiveness of their dietary efforts, while those who are more concerned about food tend to have greater optimism about the outcomes of their eating habits.

## 5. Discussion

The research aimed to explore the relationships between emotional food cravings and dieting behaviors using the Health Belief Model (HBM), with a focus on understanding the psychological and behavioral factors influencing these dynamics. Regarding data collection, convenience sampling was used to recruit participants from a subclinical population in Western Romania. The questionnaire, administered via Google Forms, was distributed through various social media groups from May to June 2024. This approach was designed to capture a diverse sample and enhance the generalizability of the findings.

In the present study, the alignment of results with the HBM is particularly evident in the relationships identified between emotional food cravings, bulimia, and other eating behaviors. According to the specialized literature, results highlight a strong connection between emotional food cravings and bulimia, as well as excessive preoccupation with food. Verzijl and collaborators emphasized the crucial role of emotional cravings in uncontrolled and emotional eating, suggesting that these cravings can trigger binge-eating episodes in individuals with bulimia [51]. Similarly, Waters, Hill, and Waller demonstrated that bulimic individuals tend to respond to food cravings in a way that reflects their emotional state rather than actual hunger [52]. These findings are further supported by evidence showing that food cravings, often amplified by emotional disorders, can perpetuate the cycle of binge-eating and compensatory behaviors characteristic of bulimia.

This complex relationship underscores the necessity for a thorough understanding of the emotional factors influencing maladaptive eating behaviors. Interventions targeting emotion management and craving reduction may be essential for the effective treatment of bulimia and the prevention of relapses. Furthermore, findings highlight the significant link between emotional food desires and culinary behavior. Previous studies have shown that emotional food cravings might predict poor short-term weight loss outcomes, underscoring the strong effect of emotions on eating patterns [53,69].

Results from the present study also highlight a significant relationship between emotional food cravings and the expectation of positive outcomes, aligning with findings in the existing literature and consistent with the HBM [54,55,56,57,58,59,60,61,62,63,64,65,66,67,68,70,71,72,73,74,75]. Specifically, research [54] found that therapies such as Emotional Freedom Techniques (EFTs) and cognitive behavioral therapy (CBT) not only lower food cravings but also enhance participants’ expectations of successful results in managing their cravings and weight. This aligns with the HBM’s construct of perceived benefits, where individuals are more likely to engage in behaviors if they anticipate positive outcomes. Furthermore, a long-term follow-up study by Sabik and collaborators [76] supports the notion that online delivery of EFT can sustain these positive expectations for up to two years, demonstrating that controlling emotional demands can foster optimism about achieving desired goals. This relationship is further reinforced by Scheier and Carver [77], who discovered that positive expectations related to appearance and self-perception can significantly impact psychological and biological stress processes. This underscores the HBM’s concept of perceived benefits, emphasizing that fostering positive expectations is crucial for managing emotional food cravings effectively.

This research also highlights the limitations of the HBM in accounting for cultural and social influences on dietary behaviors. The findings call for a theoretical shift towards models that incorporate cultural variations and social contexts, broadening the applicability of health behavior theories across diverse populations. This shift is essential for developing more comprehensive models that reflect the complex interplay of various factors influencing eating behaviors.

Furthermore, the research underscores the importance of providing educational resources that help individuals understand the interplay between emotional cravings and dieting behaviors. Support groups, workshops, and educational materials focusing on emotional well-being and its impact on eating habits can empower individuals to manage their food cravings more effectively. These resources can provide valuable support and guidance for individuals seeking to improve their eating behaviors.

The findings from this study can also inform policy and program development by emphasizing the need for policies that support mental health and emotional well-being in the context of nutrition. Public health initiatives should integrate strategies that address emotional factors and promote mental health to support healthier eating behaviors across populations. Policies that consider the emotional aspects of eating can contribute to more effective and holistic public health strategies.

Thus, the integration of these results with the HBM framework not only validates the theoretical model but also provides actionable insights for developing targeted interventions. By understanding the psychological and behavioral factors associated with emotional food cravings, the study contributes to a comprehensive approach to improving dieting behaviors and managing eating attitudes.

### 5.1. Theoretical Implications

This study’s exploration of the relationships between emotional food cravings and dieting behaviors through the lens of the HBM contributes to several theoretical advancements in the field of psycho-nutritional research. By emphasizing the role of emotional food cravings in dieting behaviors, this research extends the application of the HBM beyond its traditional scope. This integration of emotional factors enriches the theoretical framework by illustrating how emotional states influence perceptions of susceptibility, severity, and benefits related to dieting. This expansion encourages a more nuanced understanding of how emotional and psychological elements interact with health beliefs to shape behavior, thus broadening the model’s applicability.

This study introduces a novel perspective by examining how emotional food cravings mediate through sequential pathways involving bulimia, culinary behaviors, food preoccupations, and expectations of positive outcomes. This approach offers a dynamic view of behavior change processes, suggesting that the HBM can be enhanced by incorporating sequential mediation to better capture the complexity of dietary behaviors. This perspective challenges the traditional HBM framework by demonstrating the intricate interplay of emotional and behavioral factors in influencing dieting practices.

Moreover, the findings underscore the importance of self-efficacy and emotional dynamics in dietary behaviors, suggesting that the HBM’s traditional focus on rational decision-making could benefit from a deeper exploration of emotional and psychological barriers. This integration refines existing theoretical constructs and advocates for an expanded model that includes emotional and contextual factors, thereby addressing the gaps in the traditional HBM framework.

### 5.2. Practical Implications

The practical implications of this research are significant for the development of effective interventions and dietary programs. The identification of emotional food cravings as a key factor influencing dieting behaviors suggests the need for interventions that address emotional regulation and stress management. Programs incorporating techniques such as EFTs and CBT may be particularly effective in reducing cravings and improving dieting outcomes. These interventions can help individuals manage their emotional states and subsequently enhance their dietary adherence.

The study’s findings advocate for personalized dietary programs that consider individual emotional states and psychological factors. By tailoring interventions to address specific emotional triggers and psychological barriers, practitioners can enhance the effectiveness of dietary strategies and improve adherence to healthy eating practices. Personalized approaches can address unique emotional challenges and support individuals in achieving their dietary goals more effectively.

The integration of emotional and psychological factors into dietary interventions promotes a holistic approach to behavior change. This approach can lead to more comprehensive and sustainable dietary practices by addressing both cognitive and emotional dimensions of eating behaviors. Such holistic strategies are likely to be more effective in fostering long-term behavioral change and improving overall health outcomes.

### 5.3. Future Research Directions

Future research should consider several key areas to build upon the findings of this study. Longitudinal studies are needed to track how emotional food cravings and related behaviors evolve over time, providing insights into the long-term effects of emotional factors on eating patterns and dietary adherence. Additionally, exploring other theoretical models, such as Social Cognitive Theory (SCT) or the Theory of Planned Behavior (TPB), could offer a more comprehensive view of how social and cognitive factors influence eating behaviors.

Investigating the impact of cultural and socioeconomic factors on the relationship between emotional food cravings and dieting behaviors is another important direction. Understanding how different cultural backgrounds and socioeconomic statuses affect emotional eating can help tailor interventions to diverse populations. Moreover, evaluating the efficacy of various intervention strategies, such as CBT and EFT, will be crucial in determining the most effective approaches for managing emotional food cravings.

Future studies should also delve into the underlying mechanisms of emotional regulation in relation to food cravings and dieting behaviors. Adopting a biopsychosocial approach can provide a better understanding of how biological, psychological, and social factors interact to influence eating behaviors. Exploring the integration of technological tools, such as mobile apps and wearable devices, could offer innovative ways to monitor and manage emotional food cravings and dietary behaviors.

Finally, research should focus on the effect of self-efficacy on behavior change, investigating how enhancing individuals’ confidence in their ability to manage emotional cravings influences dieting behaviors. Identifying specific emotional triggers that lead to food cravings and unhealthy eating patterns will also be valuable.

### 5.4. Limitations

While the HBM has provided a valuable framework for understanding health-related behaviors in this study, several limitations are worth noting, particularly in relation to the focus on emotional food cravings and dieting behaviors.

Firstly, the HBM’s emphasis on individual perceptions of susceptibility, severity, and benefits may not fully capture the broader social and environmental factors influencing eating behaviors. The present study, which examines the interplay between emotional food cravings and various dietary attitudes, reveals that external factors such as socioeconomic constraints and access to resources significantly affect dieting behaviors. The HBM’s focus on individual decision-making might overlook these critical contextual factors that can impact the effectiveness of dietary interventions.

Secondly, the HBM lacks a detailed account of the behavioral change process over time. While the model addresses decision-making at a specific moment, it does not thoroughly explore how individuals transition through stages of behavior change. This limitation is evident in the present study, which requires an understanding of how emotional food cravings influence dieting behaviors over time. The model’s static perspective does not adequately address the dynamic nature of behavioral change as individuals evolve through different phases of dietary modification.

Additionally, the HBM framework does not sufficiently address the emotional and psychological factors that significantly influence eating behaviors. Findings highlight the substantial role of emotional responses and psychological barriers, such as stress and anxiety, in shaping food cravings and dieting practices. The model’s focus on rational decision-making processes fails to capture these complex emotional dynamics, which are crucial for understanding the nuances of eating behaviors.

Moreover, the HBM’s assumption of rational decision-making may not fully align with real-world behaviors, where irrational beliefs, habits, and unconscious processes play a significant role. For instance, despite an understanding of the benefits of healthy eating, individuals may continue unhealthy dietary practices due to emotional factors or ingrained habits. The HBM’s rational framework does not encompass these less tangible influences that affect dietary choices.

The model also falls short in considering cultural and contextual variations in health beliefs. The study indicates that cultural factors can profoundly influence food-related attitudes and behaviors, suggesting that application of the HBM may be limited when addressing diverse populations. The model’s development based on Western health perceptions may not fully account for cultural differences in dietary practices and beliefs.

Furthermore, the HBM does not adequately address the role of social influences, such as peer pressure and social support, which are significant in shaping eating behaviors. Research underscores the importance of these social factors in influencing food-related attitudes, pointing to a need for incorporating social context into behavioral models.

Lastly, while the HBM includes self-efficacy as a construct, it may not fully capture the complexities of confidence in one’s ability to change behavior. Findings suggest that self-efficacy is a critical factor in managing food cravings and modifying dieting behaviors. However, the HBM’s treatment of self-efficacy might be too simplistic to fully understand its impact on behavior change.

In conclusion, while the HBM offers valuable insights into the relationships between emotional food cravings and dieting behaviors, its limitations suggest that a more comprehensive approach may be necessary. Integrating HBM with other theoretical models that address social influences, emotional factors, and the dynamic nature of behavior change could enhance the effectiveness of interventions and provide a more holistic understanding of eating behaviors.

In addition to the theoretical limitations of the HBM, this study has several methodological constraints. One limitation is the sample size and its characteristics. The sample, while substantial, may not be fully representative of the general population, particularly regarding diversity in socioeconomic background, cultural practices, or geographic location. Future research should aim to include a more diverse sample to improve generalizability. Moreover, the data collection methods relied on self-reported measures, which are subject to biases such as social desirability and inaccurate recall. These limitations could impact the reliability of the findings, especially in a context where emotional and psychological factors are at play. Future studies could benefit from incorporating more objective measures to complement self-reports and provide a more holistic understanding of eating behaviors.

Lastly, while this study offers valuable insights into the relationships between emotional food cravings and dietary behaviors, the cross-sectional design limits the ability to infer causal relationships. Longitudinal studies are needed to better understand how emotional food cravings and eating behaviors evolve over time, providing clearer insights into the causal pathways that underlie these dynamics. Future research should also explore interventions tailored to these identified limitations, focusing on practical strategies to mitigate emotional food cravings and improve long-term eating habits.

## 6. Conclusions

This study investigated the complex dynamics between emotional food craving, psycho-nutritional mediators (bulimia tendencies, culinary behaviors, food preoccupations, and expectations of positive outcomes), and attitudes toward dieting behaviors. The findings underscore several key insights into the psycho-nutritional dynamics influencing individuals’ approaches to managing their eating behaviors.

The results revealed significant correlations and sequential mediation pathways linking emotional food craving to attitudes toward dieting behaviors. Emotional food craving was found to exert indirect effects through multiple mediators, highlighting the complex and multifaceted nature of these relationships. Specifically, higher levels of emotional food craving were associated with increased tendencies toward bulimia, heightened culinary behaviors, greater food preoccupations, and expectations of positive outcomes, all of which collectively influenced attitudes toward dieting.

This study’s findings reveal important routes by which emotional food hunger impacts attitudes toward dieting practices through intermediary psycho-nutritional variables. Higher levels of emotional food craving were consistently associated with increased tendencies toward bulimia, intensified culinary behaviors, heightened food preoccupations, and increased expectations of positive dietary outcomes. These correlations shed light on the complicated mechanisms by which emotional variables influence how people manage their diets.

## Figures and Tables

**Figure 1 healthcare-12-01934-f001:**
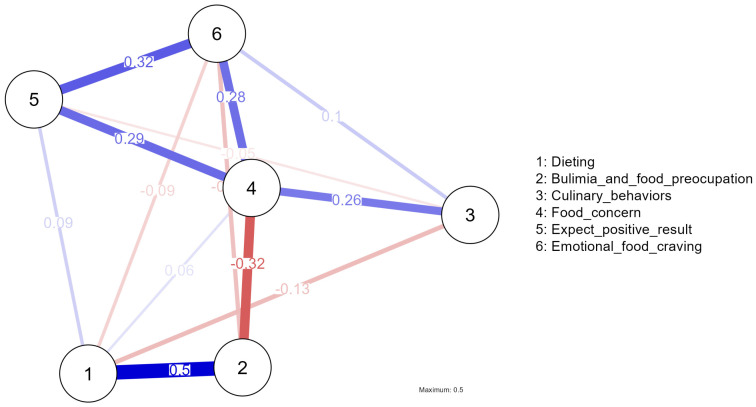
Network analysis.

**Table 1 healthcare-12-01934-t001:** Descriptive statistics of participant characteristics.

Variable	N	Minimum	Maximum	Mean	Std. Deviation	Frequency (n)	Percentage (%)
Age (years)	659	16	66	31.16	11.967		
Height (cm)	659	152.00	190.00	171.22	38.95616		
Weight (kg)	659	40.00	163.00	69.14	17.02434		
Income (RON)	657	0	15,000	3727.46	2500.628		
Gender							
-Female						535	81.2
-Male						124	18.8
Residential area							
-Urban						457	69.3
-Rural						202	30.7
Educational level							
-Secondary						340	51.6
-Higher Education						175	26.6
-Undergraduate						131	19.9
-Postgraduate						13	2.0

**Table 2 healthcare-12-01934-t002:** Descriptive statistics.

Variables–Test Used	Valid	Missing	Mean	Std. Deviation	Minimum	Maximum
1. Dieting—EAT	659	0	52.273	12.425	13.000	78.000
2. Bulimia and food preoccupation—EAT	659	0	28.821	4.889	6.000	36.000
3. Culinary behaviors—G-FCQ-T	659	0	11.457	5.802	6.000	36.000
4. Food preoccupation—G-FCQ-T	659	0	14.310	7.204	6.000	36.000
5. Expect positive result—G-FCQ-T	659	0	16.543	6.644	5.000	30.000
6. Emotional food craving—G-FCQ-T	659	0	11.364	6.320	4.000	24.000

**Table 3 healthcare-12-01934-t003:** Correlation matrix (Pearson’s correlations).

Variables	1	2	3	4	5
1. Dieting	—				
2. Bulimia and food preoccupation	0.590 ***	—			
3. Culinary behaviors	−0.288 ***	−0.309 ***	—		
4. Food preoccupation	−0.269 ***	−0.538 ***	0.431 ***	—	
5. Expecting positive results	−0.079 *	−0.250 ***	0.168 ***	0.511 ***	—
6. Emotional food craving	−0.310 ***	−0.454 ***	0.343 ***	0.592 ***	0.519 ***

* *p* < 0.05, *** *p* < 0.001.

**Table 4 healthcare-12-01934-t004:** Centrality measures per variable.

Variables	Network			
	Betweenness	Closeness	Strength	Expected Influence
1. Dieting	−0.580	−0.696	−0.020	0.123
2. Bulimia and food preoccupation	0.464	0.294	0.299	−1.463
3. Culinary behaviors	−0.580	−0.810	−1.483	−0.941
4. Food preoccupation	1.855	1.883	1.542	0.764
5. Expect positive result	−0.580	−0.307	−0.541	1.094
6. Emotional food craving	−0.580	−0.364	0.202	0.423

**Table 5 healthcare-12-01934-t005:** Sequential mediation analysis results.

Path	Coefficient (c)	SE	t-Value	*p*-Value	LLCI	ULCI
Total Effect (X → Y)	−0.6087	0.0729	−8.3448	<0.001	−0.7519	−0.4654
Direct Effect (X → Y)	−0.2330	0.0819	−2.8453	0.0046	−0.3939	−0.0722
Indirect Effect (Total)	−0.3756	0.0778			−0.5322	−0.2253
Completely Std. Indirect Effect	−0.1910	0.0386			−0.2685	−0.1152
Path	Coefficient (std.)					
Emotional food craving > Bulimia and food preoccupation	−0.4540					
Emotional food craving > Culinary behaviors	0.2559					
Emotional food craving > Food preoccupation	0.3856					
Emotional food craving > Expect positive result	0.3661					

**Table 6 healthcare-12-01934-t006:** Detailed coefficients.

Model	Variable	Coefficient	SE	*t*-Value	*p*-Value	LLCI	ULCI
Bulimia and food preoccupation	Constant	32.8116	0.3496	93.8551	<0.001	32.1251	33.4981
	Emotional food craving	−0.3512	0.0269	−13.0589	<0.001	−0.4040	−0.2984
Culinary behaviors	Constant	15.3776	1.6335	9.4139	<0.001	12.1701	18.5851
	Emotional food craving	0.2349	0.0372	6.3243	<0.001	0.1620	0.3079
	Bulimia and food preoccupation	−0.2287	0.0480	−4.7617	<0.001	−0.3230	−0.1344
Food preoccupation	Constant	19.0795	1.6912	11.2819	<0.001	15.7588	22.4003
	Emotional food craving	0.4395	0.0372	11.8205	<0.001	0.3665	0.5126
	Bulimia and food preoccupation	−0.4408	0.0475	−9.2870	<0.001	−0.5340	−0.3476
	Culinary behaviors	0.2566	0.0379	6.7643	<0.001	0.1821	0.3311
Expect positive result	Constant	4.5691	1.9071	2.3958	0.0169	0.8243	8.3140
	Emotional food craving	0.3848	0.0423	9.1054	<0.001	0.3019	0.4678
	Bulimia and food preoccupation	0.1297	0.0521	2.4899	0.0130	0.0274	0.2321
	Culinary behaviors	−0.1085	0.0405	−2.6785	0.0076	−0.1880	−0.0290
	Food preoccupation	0.3567	0.0403	8.8464	<0.001	0.2775	0.4359

## Data Availability

The original contributions presented in the study are included in the article, further inquiries can be directed to the corresponding authors.

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
