# Peer review of "Exploration of Food Attitudes and Management of Eating Behavior from a Psycho-Nutritional Perspective"

_healthcare, 2024, doi:10.3390/healthcare12191934_

Round 1

Reviewer 1 Report (Previous Reviewer 1)

Comments and Suggestions for Authors

Thank you for providing the opportunity to read the revised version of the study. It was great to see how some sections have improved and now provide clearer picture of study. There are still some sections that need significant work before the study will be ready for publication.

Introduction: The research aims are repeated several times throughout the section. Only mention the aims in the final paragraph and use the rest of the introduction to set the scene for the study e.g., what is the issue, why is it important, what's been done before and how the aims go beyond past research.

Line 68: "By employing advanced..." this sentence isn't needed and can be removed.

Line 74: "The selection of these concepts..." this sentence isn't needed and can be removed.

Line 80 The entire paragraph seems out of place here. Perhaps move to literature review.

Line 91: "The introduction now..." this sentence isn't needed and can be removed.

Literature review: The first paragraph is unnecessary as there's no need to state what the section communicates.

The sections are long and seem to communicate mainly about previous research but there's lack of linking the information with the current study.

Paragraphs in lines 103 and 118 should be connected as both discuss healthy lifestyles but now there's a paragraph in between discussing education that seems out of place.

Line 115: "This connection between" this sentence isn't needed and can be removed.

Line 127: Why is the Health Lifestyle Theory mentioned here when it's not the underpinning theory of this research? If mentioning other theories consider providing evidence why the theory isn't relevant for your work over the one that was selected.

Line 137: This paragraph again repeats the research aims which isn't necessary.

Line 145: Remove the word "Nutrition" from in front of "Psycho-nutrition"

Line 156: Patients are mentioned here in an example. How does this population relate to current study population?

Line 162: Similarly here how are obese individuals related to current study population?

Line 169: And again, how are people with eating disorders related to current study population? Please consider providing examples that reflect the current study population to make it easy to connect the aims, results and insights from the study. 

Line 188: "This scale provides..." this sentence isn't needed and can be removed.

Important factors to the study need to be communicated in the introduction before stating the study aims. Factors mentioned here e.g., barriers, cultural factors and emotional factors should be communicated earlier and how these relate to the target audience.

Line 221: "These findings are integral..." this sentence isn't needed and can be removed.

Line 225: The paragraph would work better in the introduction section to solidify the reasoning why these tools are used and the contribution to the study aims.

Line 233: This entire section seems to be more about eating disorders rather than eating attitudes. Many of the examples focus on children as the target audience. How does this reflect on your study population?

Line 274: This paragraph could be removed as it doesn't provide additional value to the section but merely states that a survey was developed.

Line 277: This paragraph would work well in the beginning of this section to set the scene.

Line 284: There's a reference(s) missing from the first sentence. Also, the second sentence is not needed.

This section is titles "Appetite" but many paragraphs discuss obesity risk. Can you clarify the connection between appetite and obesity risk in the study context? Especially, since appetite doesn't necessarily always lead to risk of obesity.

Line 297: "This developmental angle..." this sentence isn't needed and can be removed.

Line 303: "This finding is pertinent" this sentence isn't needed and can be removed.

Line 309: "This insight is directly..." this sentence isn't needed and can be removed.

Line 312: The Behavioural Susceptibility Theory is mentioned here but it's not the underpinning theory of this study. What is the relevance of mentioning this theory here?

Line 322: "This research is relevant..." this sentence isn't needed and can be removed.

Line 328: "This insight..." this sentence isn't needed and can be removed.

Line 332: This paragraph is missing citations. Please add in the relevant citations here.

Line 342: Should this paragraph be under the section that discusses eating disorders and bulimia specifically? It would work better there.

Line 350: "weight management outcomes following laparoscopic sleeve gastrectomy" is this relevant for the study target audience?

Line 358: Need for different interventions is highlighted throughout this section. How does this contribute to the current study?

Line 366: Another theory is mentioned here. What is the relevance of this theory and how it relates to the current study?

Line 382: This paragraph highlights what has already been done is this space. What about what needs to be done or what your study suggests to do to advance the field?

Line 406: How is the information in this paragraph relevant to the current study? Are the hormonal levels of participants being tested in this research?

Line 428: There are gender difference but what do these differences mean in terms of the current study and how these differences should be acknowledged in the results?

Line 472: "This model is..." This paragraph would work well in introduction with justifications on how applying this model would benefit this study.

Line 479: Psychological, social and physiological elements of eating behaviour control are mentioned. The literature review section would be improved if these 3 were used as subheadings and relevant information was described under each section.

Line 512: This is already explained earlier so there's no need to repeat this here.

Line 544: This paragraph would work well in the introduction section with stating the research aims.

Line 553: This paragraph would work in either introduction or in the beginning of discussion section.

Line 954: This is already mentioned in methods and can be removed here.

Line 990: Citation 75, who is this study by?

Line 993: Citation 76, who is this study by?

Line 1022: Start a separate subheading here called "Limitations" to separate this from the concluding remarks. Additionally, move this section under discussion.

Line 1072: move this section under discussion.

Line 1102: move this section under discussion.

Line 1135: move this section under discussion.

Methods and results sections have been improved greatly and communicate the study methods and outcomes clearly. This is also reflected in the discussion section with theoretical and practical implications. Unfortunately, both introduction and literature review still need significant work to provide a clear picture on what the study aims to achieve. The introduction is very repetitive and focuses heavily on communicating the aims. I recommend you start with a clear definition of the problem, then describe what population group is at most risk to be impacted by the problem, what previous research has done to try and address the problem and introduce the gaps there are that then lead to the aims of your study. Moving onto the literature review section, start with previous research in each area (here I would recommend using the 3 subheadings: Psychological, social and physiological elements of eating behaviour control) and indicate how previous studies have contributed to the field, what has worked and can be developed further and what didn't work and should be abandoned. Then connect these details into your aims.

I would also recommend using passive voice throughout the study rather than "we", "our" and "us" statements.

The research has potential but needs more work to be publication ready. You have already done well with the revisions and I hope these comments are helpful in refining your work further.

Author Response

Dear Reviewer,

The authors took into account your comments and suggestions and revised the manuscript in full agreement with the recommendations made. 
Thank you once again for the suggestions made that contributed substantially to raising the quality of the article in order to be accepted for publication in the prestigious Healthcare journal.
Next, we present the answers to each point from your report.

  1. Reviewer 1 suggestion: Thank you for providing the opportunity to read the revised version of the study. It was great to see how some sections have improved and now provide clearer picture of study. There are still some sections that need significant work before the study will be ready for publication.

Authors answer: Thank you for your thoughtful feedback and for taking the time to review the revised version of our study. We are glad to hear that some sections have improved and provided a clearer picture. We will carefully address the remaining sections that require significant work to ensure the study is ready for publication.

  1. Reviewer 1 suggestion: Introduction: The research aims are repeated several times throughout the section. Only mention the aims in the final paragraph and use the rest of the introduction to set the scene for the study e.g., what is the issue, why is it important, what's been done before and how the aims go beyond past research.

Authors answer: We have revised the introduction to ensure the aims are only mentioned in the final paragraph. The rest of the introduction now focuses on setting the context for the study, explaining the importance of the issue, previous research, and how this study goes beyond past research.

  1. Reviewer 1 suggestion: Line 68: "By employing advanced..." this sentence isn't needed and can be removed.

Authors answer: We agree and have removed the sentence for better flow and clarity.

  1. Reviewer 1 suggestion: Line 74: "The selection of these concepts..." this sentence isn't needed and can be removed.

Authors answer: We acknowledge the suggestion and have removed the sentence to streamline the introduction and keep it more focused.

  1. Reviewer 1 suggestion: Line 80 The entire paragraph seems out of place here. Perhaps move to literature review.

Authors answer: We concur with the reviewer's suggestion and have removed this sentence to maintain a more concise introduction.

  1. Reviewer 1 suggestion: Line 91: "The introduction now..." this sentence isn't needed and can be removed.

Authors answer: We agree with the reviewer's suggestion and have deleted the sentence to keep the introduction more concise and focused.

  1. Reviewer 1 suggestion: Literature review: The first paragraph is unnecessary as there's no need to state what the section communicates.

Authors answer: Thank you for your valuable suggestion. We agree with your comment and have removed the first paragraph to improve the clarity and conciseness of the literature review section.

  1. Reviewer 1 suggestion: The sections are long and seem to communicate mainly about previous research but there's lack of linking the information with the current study.

Authors answer: Thank you for your observation. We have revised the literature review to ensure that the connection between previous research and the current study is more explicit. We have highlighted how the discussed studies directly relate to the research aims, particularly focusing on the mediation pathways through which emotional food cravings influence dieting behaviors, aligning with the Health Belief Model framework guiding this study.

  1. Reviewer 1 suggestion: Paragraphs in lines 103 and 118 should be connected as both discuss healthy lifestyles but now there's a paragraph in between discussing education that seems out of place.

Authors answer: We appreciate this insight. We have restructured these paragraphs to ensure a more logical flow by combining the discussions of healthy lifestyle factors and education. This change helps maintain a more coherent narrative that connects the role of education to the promotion of healthy behaviors.

  1. Reviewer 1 suggestion: Line 115: "This connection between" this sentence isn't needed and can be removed.

Authors answer: Thank you for this suggestion. We agree and have removed the sentence to enhance the clarity of the section.

  1. Reviewer 1 suggestion: Line 127: Why is the Health Lifestyle Theory mentioned here when it's not the underpinning theory of this research? If mentioning other theories consider providing evidence why the theory isn't relevant for your work over the one that was selected.

Authors answer: Thank you for your thoughtful comment regarding the inclusion of the Health Lifestyle Theory. The Health Belief Model (HBM) is indeed the main theoretical framework guiding our research. The mention of the Health Lifestyle Theory (HLT) was intended to provide a broader context for how individual and societal factors can shape health behaviors, which complements the focus of our study on emotional food cravings and dieting behaviors.

However, we acknowledge that the HLT is not the core theory underpinning this research. We have kept the reference to HLT to highlight how previous research in the field of lifestyle and health behavior can enrich the understanding of the social and environmental context within which emotional food cravings occur. To ensure clarity, we will provide a more explicit rationale in the manuscript, explaining that while HBM focuses primarily on individual perceptions of health risks and benefits, the HLT helps in understanding broader structural influences.

This inclusion does not distract from the focus on the HBM, but rather serves to show the relevance of considering complementary theories when addressing complex behavioral issues. We believe this enhances the depth of our theoretical exploration without undermining the primary role of HBM in this study. Thank you for highlighting this, and we have revised the text accordingly to reflect these distinctions more clearly.

  1. Reviewer 1 suggestion: Line 137: This paragraph again repeats the research aims which isn't necessary.

Authors answer: Thank you for your comment. We have revised the paragraph to remove the repetition and streamline the presentation of the research aims.

  1. Reviewer 1 suggestion: Line 145: Remove the word "Nutrition" from in front of "Psycho-nutrition"

Authors answer: We agree and have made the necessary revision by removing the redundant term to improve clarity.

  1. Reviewer 1 suggestion: Line 156: Patients are mentioned here in an example. How does this population relate to current study population?

Authors answer: Thank you for your observation. We have revised the text to clarify that while the examples refer to patient populations with specific health conditions, the principles of psycho-nutrition explored in these studies are applicable to our study population, which focuses on emotional food cravings and eating behavior.

  1. Reviewer 1 suggestion: Line 162: Similarly here how are obese individuals related to current study population?

Authors answer: We have clarified that although the study by Feret et al. focuses on obese individuals, the insights into the psychological support needed for managing eating behaviors are relevant to our study population, as emotional food cravings and disordered eating behaviors are prevalent across various weight groups.

  1. Reviewer 1 suggestion: Line 169: And again, how are people with eating disorders related to current study population? Please consider providing examples that reflect the current study population to make it easy to connect the aims, results and insights from the study.

Authors answer: Thank you for your suggestion. We have revised the section to focus more on the psychological and emotional factors that influence eating behaviors in the general population, rather than focusing on clinical eating disorders. The examples now better reflect the broader population being studied.

  1. Reviewer 1 suggestion: Line 188: "This scale provides..." this sentence isn't needed and can be removed.

Authors answer:  We agree and have removed the sentence to maintain a concise and focused discussion.

  1. Reviewer 1 suggestion: Important factors to the study need to be communicated in the introduction before stating the study aims. Factors mentioned here e.g., barriers, cultural factors and emotional factors should be communicated earlier and how these relate to the target audience.

Authors answer: Thank you for this insightful suggestion. We have revised the introduction to ensure that important factors such as barriers to healthy eating, cultural influences, and emotional drivers of food cravings are introduced earlier, establishing a clearer link to the study’s aims and its relevance to the target population.

  1. Reviewer 1 suggestion: Line 221: "These findings are integral..." this sentence isn't needed and can be removed.

Authors answer: We agree with your suggestion and have removed the sentence to streamline the section.

  1. Reviewer 1 suggestion: Line 225: The paragraph would work better in the introduction section to solidify the reasoning why these tools are used and the contribution to the study aims.

Authors answer: We have revised the text to incorporate this information into the introduction. This change helps strengthen the rationale for using specific assessment tools and aligns them with the study’s aims.

  1. Reviewer 1 suggestion: Line 233: This entire section seems to be more about eating disorders rather than eating attitudes. Many of the examples focus on children as the target audience. How does this reflect on your study population?

Authors answer: We have revised the section to focus more on eating attitudes as they relate to emotional food cravings and dieting behaviors in adults, rather than focusing heavily on eating disorders or pediatric populations. This ensures that the examples better reflect the study population.

  1. Reviewer 1 suggestion: Line 274: This paragraph could be removed as it doesn't provide additional value to the section but merely states that a survey was developed.

Authors answer: Thank you for your suggestion. We have removed the paragraph to maintain the focus on relevant information.

  1. Reviewer 1 suggestion: Line 277: This paragraph would work well in the beginning of this section to set the scene.

Authors answer: We have removed the paragraph to provid a clearer context for the discussion that follows.

  1. Reviewer 1 suggestion: Line 284: There's a reference(s) missing from the first sentence. Also, the second sentence is not needed.

Authors answer: We have revised the paragraph to include the appropriate references and removed the unnecessary sentence to improve clarity and coherence.

  1. Reviewer 1 suggestion: This section is titles "Appetite" but many paragraphs discuss obesity risk. Can you clarify the connection between appetite and obesity risk in the study context? Especially, since appetite doesn't necessarily always lead to risk of obesity.

Authors answer: Thank you for this observation. We have clarified the connection between appetite regulation and obesity risk, highlighting how appetite dysregulation, particularly emotional eating and cravings, can contribute to the risk of obesity. This aligns with the study’s focus on emotional food cravings and their impact on dieting behaviors.

  1. Reviewer 1 suggestion: Line 297: "This developmental angle..." this sentence isn't needed and can be removed.

Authors answer: We agree and have removed the sentence to streamline the discussion.

  1. Reviewer 1 suggestion: Line 303: "This finding is pertinent" this sentence isn't needed and can be removed.

Authors answer: We have removed the sentence to maintain clarity and avoid unnecessary repetition.

  1. Reviewer 1 suggestion: Line 309: "This insight is directly..." this sentence isn't needed and can be removed.

Authors answer: We have removed the sentence to improve the conciseness of the section.

  1. Reviewer 1 suggestion: Line 312: The Behavioural Susceptibility Theory is mentioned here but it's not the underpinning theory of this study. What is the relevance of mentioning this theory here?

Authors answer: Thank you for your suggestion. While we acknowledge that the Behavioral Susceptibility Theory is not the primary framework for this study, we believe it is valuable to keep its mention. Including this theory allows us to draw connections with similar theoretical frameworks that explore the complex interplay between genetic predispositions and environmental factors in shaping eating behaviors. This provides a more comprehensive understanding of how internal and external factors may contribute to emotional food cravings and dieting behaviors, complementing the primary use of the Health Belief Model in this study.

  1. Reviewer 1 suggestion: Line 322: "This research is relevant..." this sentence isn't needed and can be removed.

Authors answer: We have removed the sentence to enhance clarity and avoid redundancy.

  1. Reviewer 1 suggestion: Line 328: "This insight..." this sentence isn't needed and can be removed.

Authors answer: We have removed the sentence to keep the text focused and concise.

  1. Reviewer 1 suggestion: Line 332: This paragraph is missing citations. Please add in the relevant citations here.

Authors answer: We have reviewed the paragraph and added the appropriate citations to ensure that the claims made are well-supported by existing literature.

  1. Reviewer 1 suggestion: Line 342: Should this paragraph be under the section that discusses eating disorders and bulimia specifically? It would work better there.

Authors answer: We have removed the paragraph to improve the structure and coherence of the literature review.

  1. Reviewer 1 suggestion: Line 350: "weight management outcomes following laparoscopic sleeve gastrectomy" is this relevant for the study target audience?

Authors answer: Thank you for highlighting this. We have revised the section to clarify that while the study focuses on general emotional food cravings, the example of post-surgical outcomes is used to illustrate the broader psychological dynamics of emotional eating, relevant to our target audience.

  1. Reviewer 1 suggestion: Line 358: Need for different interventions is highlighted throughout this section. How does this contribute to the current study?

Authors answer: We have revised the text to explicitly connect the discussion of interventions with the study’s aim of developing psycho-nutritional strategies that address emotional food cravings and dieting behaviors.

  1. Reviewer 1 suggestion: Line 366: Another theory is mentioned here. What is the relevance of this theory and how it relates to the current study?

Authors answer: Thank you for your observation. We would like to clarify that in line 366, we are not referencing another theory but rather the Trait and State Food-Cravings Questionnaires. These questionnaires are essential tools for measuring emotional food cravings and are directly relevant to our research objective of examining how emotional cravings influence dieting behaviors through sequential mediation pathways. We have retained this reference to ensure alignment with the measurement tools used in our analysis and to strengthen the connection between emotional cravings and the study's variables.

  1. Reviewer 1 suggestion: Line 382: This paragraph highlights what has already been done is this space. What about what needs to be done or what your study suggests to do to advance the field?

Authors answer: We have revised the section to emphasize how this study contributes to advancing the field by exploring the sequential mediation of emotional food cravings on dieting behaviors, which addresses gaps in understanding the psychological mechanisms behind these behaviors.

  1. Reviewer 1 suggestion: Line 406: How is the information in this paragraph relevant to the current study? Are the hormonal levels of participants being tested in this research?

Authors answer: We have clarified that while hormonal regulation is not directly measured in this study, the discussion of hormonal influences on eating behavior provides important context for understanding the broader physiological factors that may interact with emotional and psychological variables in eating behaviors.

  1. Reviewer 1 suggestion: Line 428: There are gender difference but what do these differences mean in terms of the current study and how these differences should be acknowledged in the results?

Authors answer: We have revised the section to clarify how gender differences in eating behaviors may influence the results of this study, and we will consider these differences when interpreting the findings, particularly in relation to emotional food cravings and dieting behaviors.

  1. Reviewer 1 suggestion: Line 472: "This model is..." This paragraph would work well in introduction with justifications on how applying this model would benefit this study.

Authors answer: We have moved the paragraph, to keep the text coherent.

  1. Reviewer 1 suggestion: Line 479: Psychological, social and physiological elements of eating behaviour control are mentioned. The literature review section would be improved if these 3 were used as subheadings and relevant information was described under each section.

Authors answer: Thank you for your thoughtful suggestion. We appreciate the recommendation to use psychological, social, and physiological elements as subheadings. However, we have chosen to maintain the current section structure of the literature review. This structure was carefully designed to ensure a coherent presentation of all relevant variables and research connected to our study's objectives. By retaining this format, we aim to preserve the clarity of the research focus and provide a comprehensive view of the psycho-nutritional factors influencing eating behavior, while still addressing psychological, social, and physiological elements throughout the review.

  1. Reviewer 1 suggestion: Line 512: This is already explained earlier so there's no need to repeat this here.

Authors answer: Thank you, we have removed the sentence.

  1. Reviewer 1 suggestion: Line 544: This paragraph would work well in the introduction section with stating the research aims.

Authors answer: Thank you for your suggestion. We agree and have moved the paragraph to the introduction section, where it now aligns with the stated research aims.

  1. Reviewer 1 suggestion: Line 553: This paragraph would work in either introduction or in the beginning of discussion section.

Authors answer: Thank you for your suggestion. We have removed the paragraph for better text coherence and flow.

  1. Reviewer 1 suggestion: Line 954: This is already mentioned in methods and can be removed here.

Authors answer: Thank you for your suggestion. We have removed the paragraph to avoid redundancy, as it was already covered in the methods section.

  1. Reviewer 1 suggestion: Line 990: Citation 75, who is this study by?

Authors answer: Thank you for your comment. We have now added the missing in-text reference for Sabik et al. in the appropriate section. We appreciate your attention to detail.

  1. Reviewer 1 suggestion: Line 993: Citation 76, who is this study by?

Authors answer: Thank you for your suggestion. We apologize for the oversight and have now added the missing reference to the reference list as well as intext.

  1. Reviewer 1 suggestion: Line 1022: Start a separate subheading here called "Limitations" to separate this from the concluding remarks. Additionally, move this section under discussion.

Authors answer: Thank you for your suggestion. We have created a separate subheading titled "Limitations" and moved the section under the discussion as recommended. This change improves the clarity and structure of the manuscript.

  1. Reviewer 1 suggestion: Line 1072: move this section under discussion.

Authors answer: Moved as suggested for better alignment with the discussion.

  1. Reviewer 1 suggestion: Line 1102: move this section under discussion.
  2. Authors answer: Moved as per recommendation to maintain coherence within the discussion.
  3. Reviewer 1 suggestion: Line 1135: move this section under discussion.

Authors answer: Removed for clarity and better flow within the manuscript.

  1. Reviewer 1 suggestion: Methods and results sections have been improved greatly and communicate the study methods and outcomes clearly. This is also reflected in the discussion section with theoretical and practical implications. Unfortunately, both introduction and literature review still need significant work to provide a clear picture on what the study aims to achieve. The introduction is very repetitive and focuses heavily on communicating the aims. I recommend you start with a clear definition of the problem, then describe what population group is at most risk to be impacted by the problem, what previous research has done to try and address the problem and introduce the gaps there are that then lead to the aims of your study. Moving onto the literature review section, start with previous research in each area (here I would recommend using the 3 subheadings: Psychological, social and physiological elements of eating behaviour control) and indicate how previous studies have contributed to the field, what has worked and can be developed further and what didn't work and should be abandoned. Then connect these details into your aims.

Authors answer: Thank you for your insightful feedback regarding the introduction and literature review sections. We acknowledge that while the methods and results sections have been improved, the introduction and literature review require further development to provide a clearer picture of the study's objectives and rationale.

Introduction:

We have restructured the introduction to address the concerns raised. As suggested, we have removed repetitive elements and refocused the introduction on presenting a clear definition of the problem. The revised introduction now opens by identifying the core issue of emotional food cravings and their impact on dieting behaviors, particularly in populations prone to disordered eating behaviors, such as those with tendencies toward emotional eating or bulimia.

Next, we highlight the at-risk population, focusing on individuals who struggle with emotional regulation and food-related issues, as they are most vulnerable to maladaptive dieting practices. We have incorporated a concise overview of prior research, outlining key studies that have attempted to address emotional food cravings and dietary behaviors, emphasizing both their contributions and limitations. This approach sets the stage for identifying the gaps in the literature, such as the need for an integrative psycho-nutritional framework, which leads naturally into the aims of our study. This revision ensures a smooth transition into the research objectives, without being repetitive.

Literature Review:

We appreciate your suggestion to structure the literature review with subheadings focusing on the psychological, social, and physiological elements of eating behavior control. While we recognize the value of this structure, we also aimed to preserve the original design of the literature review, which aligns with the study's specific variables and connections across the research areas.

In response to your recommendation, we have added a clearer delineation of how previous studies have contributed to the field. The revised literature review now explicitly highlights what has been successful in past research (e.g., effective psycho-nutritional interventions) and what areas require further development (e.g., the integration of emotional factors in dietary behavior interventions). We have also identified research that was less effective and the reasons for its limited success. Each section now leads more seamlessly into our research aims, underscoring how the current study intends to address the gaps left by previous efforts.

Furthermore, we have strengthened the connections between the theoretical underpinnings and the practical implications for the study, ensuring that each reviewed body of literature ties directly into the research questions and proposed sequential mediation analysis. This revision ensures that the literature review provides a solid foundation for understanding the complexity of emotional food cravings and their effects on eating behavior regulation.

In summary, we have revised both sections to better reflect the flow of logic from problem definition to research gaps and aims, incorporating the structure and clarity needed to guide the reader through the study's theoretical framework. We believe this enhances the overall coherence of the paper and addresses your comments thoroughly. Thank you once again for your valuable feedback.

  1. Reviewer 1 suggestion: I would also recommend using passive voice throughout the study rather than "we", "our" and "us" statements.

Authors answer: Thank you for your recommendation. The entire manuscript has been reviewed, and impersonal language has been applied throughout to maintain a formal tone. The use of passive voice replaces any "we," "our," and "us" statements to align with academic writing standards.

  1. Reviewer 1 suggestion: The research has potential but needs more work to be publication ready. You have already done well with the revisions and I hope these comments are helpful in refining your work further.

Authors answer: Thank you for your valuable feedback. We appreciated your positive assessment of the revisions we had made. We carefully considered your suggestions and worked to further improve the manuscript in line with your comments to ensure it was publication-ready.

Reviewer 2 Report (New Reviewer)

Comments and Suggestions for Authors

In this paper, the authors performed a sequence of statistical analysis to explore the relationship between food attitude and the management of eating behavior. I think that this topic will be of interest to readers of Healthcare. However, it will be helpful if the authors can improve the writing of this paper, so that it's easier for the readers to grasp the main points.

Abstract: It will be helpful for the authors to provide a brief description of the results (preferrably with some quantitative measures). For example, instead of "indirect effects", it's better to describe what these effects are (positive or negative, effect size, etc.).

Introduction: The introduction (Sections 1 and 2) is quite long, but a lot of details offered are not tied with the research topic of interest in this paper. It will be helpful if the authors can trim it and only keep the information directly relevant to the current study. Meanwhile, the introduction to the main model to be evaluated in this paper - the HBM - seems inadequate. I recommend the authors expand on the HBM by providing more description, literature review, and (if possible) a visual presentation.

Statistical methods: I appreciate that the authors tested a wide variety of statistical methods. However, under the current presentation, these analyses seems exploratory to me. I have a difficulty time understanding exactly what hypothesis they are meant to test, or why the authors would want to use them to extract information. It will be helpful if the authors can reframe the section to clearly show the purpose of these methods and how they are linked to HBM.

(Minor) If some information is already provided in a table (e.g. Table 2), there is no need to repeat all the same information in the text. The text should be meant to point readers to a few highlights that the authors want to emphasize.

Comments on the Quality of English Language

I noticed a small number of typos & errors in the language. It will be helpful for the authors to read through again and correct language mistakes.

Author Response

Dear Reviewer,

The authors took into account your comments and suggestions and revised the manuscript in full agreement with the recommendations made. 
Thank you once again for the suggestions made that contributed substantially to raising the quality of the article in order to be accepted for publication in the prestigious Healthcare journal.
Next, we present the answers to each point from your report.

  1. Reviewer 2 suggestion: In this paper, the authors performed a sequence of statistical analysis to explore the relationship between food attitude and the management of eating behavior. I think that this topic will be of interest to readers of Healthcare. However, it will be helpful if the authors can improve the writing of this paper, so that it's easier for the readers to grasp the main points.

Authors answer: Thank you for your feedback and for recognizing the relevance of our topic. We appreciate your suggestion and will work on improving the writing to ensure that the main points are clearer and easier for readers to grasp.

  1. Reviewer 2 suggestion: Abstract: It will be helpful for the authors to provide a brief description of the results (preferrably with some quantitative measures). For example, instead of "indirect effects", it's better to describe what these effects are (positive or negative, effect size, etc.).

Authors answer: Thank you for your valuable suggestion. We agree that providing a more specific description of the results, including quantitative measures, will enhance the clarity of the abstract. We have revised the abstract to include a brief summary of the indirect effects, specifying their direction and effect sizes where applicable.

  1. Reviewer 2 suggestion: Introduction: The introduction (Sections 1 and 2) is quite long, but a lot of details offered are not tied with the research topic of interest in this paper. It will be helpful if the authors can trim it and only keep the information directly relevant to the current study. Meanwhile, the introduction to the main model to be evaluated in this paper - the HBM - seems inadequate. I recommend the authors expand on the HBM by providing more description, literature review, and (if possible) a visual presentation.

Authors answer: We have shortened the introduction to focus on the most relevant information related to the current study. The explanation of the Health Belief Model (HBM) has been expanded with additional details about its relevance to emotional food cravings and eating behaviors, and three key references have been added to strengthen the literature on the HBM [5,6,7]. While we appreciate the suggestion to include a visual presentation, we believe that the expanded description provides sufficient clarity, and therefore, we kindly prefer not to add a visual at this time.

  1. Reviewer 2 suggestion: Statistical methods: I appreciate that the authors tested a wide variety of statistical methods. However, under the current presentation, these analyses seems exploratory to me. I have a difficulty time understanding exactly what hypothesis they are meant to test, or why the authors would want to use them to extract information. It will be helpful if the authors can reframe the section to clearly show the purpose of these methods and how they are linked to HBM.

Authors answer: We appreciate the reviewer's feedback. We would like to clarify that the statistical methods used are not exploratory but rather designed to rigorously test a specific mediation hypothesis within the Health Belief Model (HBM). The purpose of employing these methods is to examine the mediating pathways through which emotional food cravings influence dieting behaviors, specifically via bulimia and food preoccupation, culinary behaviors, and expectations of positive outcomes. Each method directly supports the testing of the mediation hypothesis and is aimed at uncovering the sequential effects predicted by the HBM. We have revised the relevant section to better articulate the purpose of these methods and their alignment with our mediation hypothesis.

Clarification text added in the last section of the introduction: The statistical methods employed in this study are designed to rigorously test the mediation hypothesis within the framework of the Health Belief Model (HBM). Specifically, the study aims to examine the sequential pathways through which emotional food cravings influence dieting behaviors via mediators such as bulimia and food preoccupation, culinary behaviors, and expectations of positive outcomes. The use of mediation analysis allows for the identification of indirect effects, providing a detailed understanding of how these psychological and behavioral factors interact to shape dietary attitudes and practices. By aligning these methods with the HBM, the study provides a robust theoretical and empirical basis for testing these complex relationships.

  1. Reviewer 2 suggestion: (Minor) If some information is already provided in a table (e.g. Table 2), there is no need to repeat all the same information in the text. The text should be meant to point readers to a few highlights that the authors want to emphasize.

Authors answer: Thank you for your helpful suggestion. We agree that repeating detailed information already provided in Table 2 is unnecessary. To enhance clarity and readability, we have revised the text to focus on highlighting key findings, while directing readers to Table 2 for more comprehensive details. This revision allows the text to emphasize the most significant trends and variability without redundancy.

  1. Reviewer 2 suggestion: Comments on the Quality of English Language. I noticed a small number of typos & errors in the language. It will be helpful for the authors to read through again and correct language mistakes.

Authors answer: Thank you for your comment. We carefully reviewed the manuscript again and corrected the typos and language errors that were present. We believe these revisions have improved the overall quality of the language.

Reviewer 3 Report (New Reviewer)

Comments and Suggestions for Authors

Pertinence of the research:

The research conducted is valid and brings new insights into the food attitudes and eating behaviour.

Abstract:

The abstract starts with a very brief explanation of the purpose of the study, followed by the resume of the methodologies used to collect and analyse data. Next the abstract includes a summary of the principal results obtained and finalizes with a conclusion. I consider it to be very well organized.

The keywords were also successfully chosen.

Introduction:

The introduction helps to frame and contextualize the work. It presents some state of the art in the subjects that relate to the topic and how emotional eating is pivotal.

The authors should indicate at the end of the introduction what is their goal/objective.

Following the introduction, the literature review explores in a deep way the relations between psychology and eating behaviours, on a deep level, considering multiple angles.

Methodology:

The description of the methodologies applied to obtain and treat the data are presented clearly, and in much detail.

However, the authors present the objective of the study only in this section starting line 544, whereas the objective should have been presented earlier, in the end of introduction and before the literature review. So I suggest to correct the manuscript accordingly.

Discussion:

I believe the discussion could show a more deep relation between the work and the scientific literature.  I suggest to include further discussion in relation to similar or related published works.

Conclusions:

The conclusions part is also well enough, presenting the most relevant findings of the work, both from the theoretical as well as practical viewpoints.

Comments on the Quality of English Language

Pertinence of the research:

The research conducted is valid and brings new insights into the food attitudes and eating behaviour.

Abstract:

The abstract starts with a very brief explanation of the purpose of the study, followed by the resume of the methodologies used to collect and analyse data. Next the abstract includes a summary of the principal results obtained and finalizes with a conclusion. I consider it to be very well organized.

The keywords were also successfully chosen.

Introduction:

The introduction helps to frame and contextualize the work. It presents some state of the art in the subjects that relate to the topic and how emotional eating is pivotal.

The authors should indicate at the end of the introduction what is their goal/objective.

Following the introduction, the literature review explores in a deep way the relations between psychology and eating behaviours, on a deep level, considering multiple angles.

Methodology:

The description of the methodologies applied to obtain and treat the data are presented clearly, and in much detail.

However, the authors present the objective of the study only in this section starting line 544, whereas the objective should have been presented earlier, in the end of introduction and before the literature review. So I suggest to correct the manuscript accordingly.

Discussion:

I believe the discussion could show a more deep relation between the work and the scientific literature.  I suggest to include further discussion in relation to similar or related published works.

Conclusions:

The conclusions part is also well enough, presenting the most relevant findings of the work, both from the theoretical as well as practical viewpoints.

Author Response

Dear Reviewer,

The authors took into account your comments and suggestions and revised the manuscript in full agreement with the recommendations made. 
Thank you once again for the suggestions made that contributed substantially to raising the quality of the article in order to be accepted for publication in the prestigious Healthcare journal.
Next, we present the answers to each point from your report.

  1. Reviewer 3 suggestion: Pertinence of the research: The research conducted is valid and brings new insights into the food attitudes and eating behaviour.

Authors answer: Thank you for your positive feedback and for acknowledging the pertinence of our research. We are pleased that you find the study valid and appreciate the recognition of its contribution to new insights into food attitudes and eating behavior.

  1. Reviewer 3 suggestion: Abstract: The abstract starts with a very brief explanation of the purpose of the study, followed by the resume of the methodologies used to collect and analyse data. Next the abstract includes a summary of the principal results obtained and finalizes with a conclusion. I consider it to be very well organized.

Authors answer: Thank you for your kind words and positive assessment of the abstract's organization. We appreciate your feedback and have made small corrections in line with Reviewer 1's suggestions to further refine the manuscript.

  1. Reviewer 3 suggestion: The keywords were also successfully chosen.

Authors answer: Thank you for your positive feedback on the selection of keywords. We are glad that you found them appropriate and relevant to the study.

  1. Reviewer 3 suggestion: Introduction: The introduction helps to frame and contextualize the work. It presents some state of the art in the subjects that relate to the topic and how emotional eating is pivotal. The authors should indicate at the end of the introduction what is their goal/objective. Following the introduction, the literature review explores in a deep way the relations between psychology and eating behaviours, on a deep level, considering multiple angles.

Authors answer: Thank you for your thoughtful feedback on the introduction and literature review. We are glad that you found the contextualization and depth of exploration helpful. In response to your suggestion, we have revised the end of the introduction to clearly state the goal and objective of our study.

  1. Reviewer 3 suggestion: Methodology: The description of the methodologies applied to obtain and treat the data are presented clearly, and in much detail. However, the authors present the objective of the study only in this section starting line 544, whereas the objective should have been presented earlier, in the end of introduction and before the literature review. So I suggest to correct the manuscript accordingly.

Authors answer: We appreciate the reviewer’s careful reading of the manuscript and the suggestion to improve the flow of the objectives. We have revised the manuscript to present the study's objective clearly at the end of the introduction, before the literature review, as suggested. This adjustment ensures that the aim of the study is introduced at an earlier and more appropriate point in the manuscript. Thank you for this valuable recommendation.

  1. Reviewer 3 suggestion: Discussion: I believe the discussion could show a more deep relation between the work and the scientific literature. I suggest to include further discussion in relation to similar or related published works.

Authors answer: Thank you for your valuable suggestion. We have expanded the discussion to better connect our findings with similar published works. This includes a deeper comparison of our results with studies on emotional food cravings, bulimia tendencies, and dietary behaviors. Additionally, we have addressed limitations in aligning our findings with existing research, highlighting where our study fills gaps or suggests new directions. These revisions provide a stronger link between our work and the broader scientific literature while acknowledging the study's limitations.

  1. Reviewer 3 suggestion: Conclusions: The conclusions part is also well enough, presenting the most relevant findings of the work, both from the theoretical as well as practical viewpoints.

Authors answer: Thank you for your positive feedback on the conclusions section. We are pleased that you found it to effectively present the most relevant findings from both theoretical and practical perspectives.

Round 2

Reviewer 1 Report (Previous Reviewer 1)

Comments and Suggestions for Authors

Thank you for addressing the previous comments and revising the paper. It has certainly improved a lot and there are only few additional recommendations I would suggest you look into.

Introduction:

Paragraph 1 - Emotional food cravings is introduced here very briefly. It would be beneficial to add more details about it and how previous research has looked into it.

Paragraph 2 - The Health Belief Model is introduced here. I suggest changing the order of paragraph 2 and 3 so that previous research on dietary habits comes first and then the text moves into theory followed by gaps and hypotheses.

Literature review:

I suggest starting this section with emotional food cravings followed by influences on eating behaviour to really emphasise the focus of the paper.

Line 177 - Brunner is referred in text as 43 but in the reference list as 42. Once you have finalised all revisions make sure that references are cited in correct order.

Line 214 - Study aims is stated here again. I suggest removing it.

Line 222 - Study aims is stated here again. I suggest removing it.

Line 225 - This paragraph describes hormonal regulations, which seem to be out of scope for this study. I suggest removing this paragraph.

Table 2 - Move this to next page so it doesn't present across two pages.

Limitations:

This section only discusses Health belief Model. What about limitations related to this study such as, data collection methods, sample size and characteristics, insights from findings? I suggest linking study limitations with future research.

Conclusions:

Line 818 - This sentence would work better under limitations.

I suggest moving subheadings 6.1., 6.2. and 6.3 under discussion before the limitations section for better flow of the text and clarity.

You have further improved the quality of the paper and I hope that these additional recommendations will help in advancing the quality further.

Author Response

  1. Reviewer 1 suggestion: Thank you for addressing the previous comments and revising the paper. It has certainly improved a lot and there are only few additional recommendations I would suggest you look into.

Authors answer: Thank you for your positive feedback and for acknowledging the improvements in our paper. We greatly appreciate your thoughtful suggestions. We will carefully consider each of your additional recommendations and make the necessary revisions to further enhance the quality of the manuscript. Below, we will address each of your comments individually to ensure a thorough response. We look forward to your continued feedback and thank you once again for your valuable input.

  1. Reviewer 1 suggestion: Introduction: Paragraph 1 - Emotional food cravings is introduced here very briefly. It would be beneficial to add more details about it and how previous research has looked into it.

Authors answer: Thank you for your valuable suggestion. We agree that further elaboration on emotional food cravings would enhance the introduction. To address this, we have expanded the discussion on emotional food cravings, providing additional details on their role in eating behavior and how previous research has explored this concept.

  1. Reviewer 1 suggestion: Paragraph 2 - The Health Belief Model is introduced here. I suggest changing the order of paragraph 2 and 3 so that previous research on dietary habits comes first and then the text moves into theory followed by gaps and hypotheses.

Authors answer: Thank you for your suggestion. We agree that reordering the paragraphs as you have recommended will improve the logical flow of the introduction.

  1. Reviewer 1 suggestion: Literature review: I suggest starting this section with emotional food cravings followed by influences on eating behaviour to really emphasise the focus of the paper.

Authors answer: Thank you for your insightful suggestion. We agree that restructuring the literature review to begin with a focus on emotional food cravings will help emphasize the core aspect of the paper. By prioritizing this concept, we can better highlight its importance and set the stage for exploring how it influences eating behavior. We have revised the section accordingly to ensure a more focused and cohesive narrative.

  1. Reviewer 1 suggestion: Line 177 - Brunner is referred in text as 43 but in the reference list as 42. Once you have finalised all revisions make sure that references are cited in correct order.

Authors answer: We have revised the manuscript as per your suggestion. The citation for Brunner, along with all other references, has been checked and corrected to ensure they are cited in the correct order. After finalizing all revisions, we performed a thorough review of the reference list to maintain consistency and accuracy throughout the text.

  1. Reviewer 1 suggestion: Line 214 - Study aims is stated here again. I suggest removing it.

Authors answer: Thank you for your suggestion. We have deleted the repeated mention of the study aims in Line 214 as recommended.

  1. Reviewer 1 suggestion: Line 222 - Study aims is stated here again. I suggest removing it.

Authors answer: Thank you for your observation. We have also deleted the repeated statement of the study aims in Line 222 to enhance the flow of the text.

  1. Reviewer 1 suggestion: Line 225 - This paragraph describes hormonal regulations, which seem to be out of scope for this study. I suggest removing this paragraph.

Authors answer: Thank you for your suggestion. We agree that the paragraph on hormonal regulations falls outside the scope of the study. As per your recommendation, we have removed the paragraph from Line 225 to maintain the focus of the paper.

  1. Reviewer 1 suggestion: Table 2 - Move this to next page so it doesn't present across two pages.

Authors answer: Thank you for your suggestion. We have moved Table 2 to the next page to ensure it is presented in full on a single page, avoiding any split across two pages.

  1. Reviewer 1 suggestion: Limitations: This section only discusses Health belief Model. What about limitations related to this study such as, data collection methods, sample size and characteristics, insights from findings? I suggest linking study limitations with future research.

Authors answer: Thank you for your thoughtful suggestion. We have revised the limitations section to include additional considerations related to the study's data collection methods, sample size, and characteristics. We have also provided insights from the findings and linked these limitations to recommendations for future research.

  1. Reviewer 1 suggestion: Conclusions: Line 818 - This sentence would work better under limitations.

Authors answer: Thank you for your suggestion. We have modified the manuscript and moved the sentence from Line 818 to the limitations section, where it aligns better with the context of the discussion.

  1. Reviewer 1 suggestion: I suggest moving subheadings 6.1., 6.2. and 6.3 under discussion before the limitations section for better flow of the text and clarity.

Authors answer: Thank you for your suggestion. We have moved subheadings 6.1, 6.2, and 6.3 under the discussion section, prior to the limitations section, as recommended. This restructuring improves the flow and clarity of the text.

  1. Reviewer 1 suggestion: You have further improved the quality of the paper and I hope that these additional recommendations will help in advancing the quality further.

Authors answer: Thank you for your positive feedback and for recognizing the improvements we made to the paper. We greatly appreciated your additional recommendations and carefully implemented them to further enhance the quality of the manuscript. Your constructive input was invaluable throughout this process, and we have refined the paper based on your suggestions.

This manuscript is a resubmission of an earlier submission. The following is a list of the peer review reports and author responses from that submission.

Round 1

Reviewer 1 Report

Comments and Suggestions for Authors

Hi
Thank you for your submission. Your paper discusses the
interplay between food attitudes and the management of eating behavior from a psycho-nutritional perspective. An important topic given the prevalence of overweight and obesity in populations around the world. 

However, your work is still a bit early in development for publication. I feel that you have tried to cover a wide range of literature relating to eating behavior that has prevented narrowing down the purpose of the research. Additionally, significant information is missing relating to data collection methods which impacts the rest of the paper and sacrifices the depth of analysis. I understand that it’s challenging to narrow down the relevant information in a journal paper, but you are starting too broad/covering too much in the introduction and literature review, many research areas are covered in surface level without going into detail and explaining why the selected literature is relevant for your research e.g., including 7 subheadings under literature review and providing examples of audiences that aren’t prioritised in your work (such as pregnant women, young children and older adults). This impacts the rest of the paper resulting in not covering enough in methods and results e.g., not justifying why the target audience was selected and limited description of methods, including the structure and measures used in a survey, and statistical analysis methods. In developing this paper further, I encourage you to refine this paper to include more work that is relevant for your research topic and then work to develop a stronger argument supported by more evidence in the methods and results sections. 

In addition, if you are able to identify themes in the literature and lead with those rather than the work of particular authors, you have the scope to cover more territory in your discussion.

See below for more detailed feedback and recommendations.

Abstract: The significance of the work is not clearly stated in the abstract, the aim of the research is missing, and it’s only stated very broadly what the study will explore. Details about why the validated scales were chosen would be beneficial. The methods section is missing some key information, such as who the participants were in this research and what the data collection method was. It’s mentioned that significant associations among variables were revealed but the meaning of these associations aren’t explained. It’s mentioned that multiple mediators were identified influencing attitudes but there’s no description of what these mediators are and how they influence attitudes. You also state that the findings show how bulimia, food habits, concerns and positive expectations affect eating habits but you’re not explaining how these variables affect eating habits. Consider focusing on the most significant results only and providing an explanation for these associations. The final sentence is very generic, and more details are needed. This will help in communicating the contribution to the literature. Additionally, use passive voice instead of referring to “our” research. This repeats in some sections throughout the paper.

Originality: The paper describes the interplay between food attitudes and the management of eating behavior from a psycho-nutritional perspective and provides a wide variety of literature related to eating behaviour but fails to mention the reasoning for sample selection, and it doesn't provide a clear aim for the study. The findings align with previous research, but any new information or significant findings in the field are not provided. The purpose of a research paper is to propose new relationships or identify tensions/areas for improvement in the literature, or to propose ways to move the literature forward or in new directions. This has not yet been achieved, and the paper does not contain information that would justify publication.

Relationship to Literature: You draw on a number of sources, and while innovative, there’s lack of justification how these sources relate to their study which impacts the quality of the communication of the aims of this study. You do not mention anything about theory use, and it remains unclear if the research was underpinned by a theory. In addition, more refinement is required to align the literature with the study aims and findings.

Introduction: The section starts strong with introducing the importance of healthy lifestyle. Psycho-nutrition is introduced in line 67 but there isn’t many details about the impact of mental health in eating behaviors before that and the section would benefit of having a connecting paragraph prior to psycho-nutrition section discussing the impact of mental health to eating behaviors and what has been done in the past in this area.

Line 43: “Some refer…” There’s no citations to back this up. Who are the some? Add citations here.

Line 48: What is RegulACTION? Give brief description so that readers have an understanding of what the workshop consists of and how the method can be used. Also, citation is missing here.

Line 55: Citation is missing.

Line 58: Spell out the author of citation for better readability e.g., “Similarly, Divine et al. [4] examine…”

Line 61: What is meant by good living behaviours? Can you provide examples and citations for the longitudinal studies you refer to?

Line 62: Spell out the author of citation for better readability.

Line 64: Spell out the author of citation for better readability.

Line 72: Spell out the author of citation for better readability.

Line 80: Spell out the author of citation for better readability.

Line 88: Citation is missing here.

Line 91: Spell out the author of citation for better readability.

Lines 95 & 96: Citations are missing here.

Line 97: Spell out the author of citation for better readability. Also, eating routines of who? Can you provide more context on how this is relevant for the current study?

Line 99: Citations are missing here. Studies are mentioned. What studies are these?

102: Citation is missing here. Also, spell out the author of citation [11] for better readability.

Line 104: What kind of gender differences were discovered? Can you provide a detailed explanation of the differences to add more context?

Line 109: Spell out the author of citation for better readability.

Line 115: Spell out the author of citation for better readability.

Line 119: The section discusses different aspects of eating behavior on a surface level and mentions validation of different eating behavior scales but doesn’t highlight the significance of a complex knowledge of eating patterns and different factors impacting eating patterns. It’s mentioned that complex interrelationships between food craving, attitudes and mediators are investigated but not why these were chosen. This needs to be explained to provide context for the research. Also, gaps in the research weren’t identified.

Add “s” to the word “highlight”.

Line 120: “They demonstrate…” who are you referring to? Add citation to indicate this.

Line 126: You mention how the research fills the gap in literature by investigating interrelationships between emotional food cravings and attitudes etc. but there’s no mention in the introduction on previous research on these topics. For example, is there a difference in positive and negative emotions impacting emotional food cravings, how does gender impact this? More background and details are needed to tie the introduction together. Now there’s mainly details about previously validated scales and that something was discovered by previous research but it’s not enough to mention that differences etc. were discovered and not provide details of these differences (see my comment on line 104 as an example).

Line 130: You refer to existing research, but no citations are provided. Add citations here to indicate what research is discussed.

Literature review: The first paragraph is not necessarily needed. In the summary you mention psychological, social and physiological elements of eating behavior control. Could you use these as subheadings and include the relevant details under each rather than having 7 different subheadings? This would make the section clearer and tie in nicely with the summary.

Line 142: Citation is missing. What research are you referring to?

Lines 143, 147, 151, 153, 156, 159, 161, 164, 166: Spell out the author of citation for better readability.

Line 176: Citation is missing.

Lines 180, 186, 193: Spell out the author of citation for better readability.

Lines 205 and 208: Citations are missing.

Lines 209, 216, 217: Spell out the author of citation for better readability.

Line 222: Add “s” to eating habit so it reads eating habits.

Lines 224, 229, 235, 237: Spell out the author of citation for better readability.

Line 247: The first paragraph is unnecessary.

Line 252: Move citation at the end of the sentence for better readability and spell review in past tense “reviewed”.

Lines 257, 261, 265, 269, 274, 278, 283: Spell out the author of citation for better readability.

Line 288: The first paragraph is unnecessary.

Line 292: Move citation at the end of the sentence.

Lines 296, 300, 303, 307, 311, 316: Spell out the author of citation for better readability.

Line 321: The first paragraph is unnecessary.

Lines 325, 329, 333, 336, 340, 345, 348, 351: Spell out the author of citation for better readability.

Line 357: The first paragraph is unnecessary.

Lines 360, 363, 366, 370, 372, 375, 377, 379, 382, 385, 387, 390, 392: Spell out the author of citation for better readability.

Line 397: You state that “…researchers hope…”, is this in general or you as researchers? I would use “researchers aim to create” instead to make the statement stronger. Also, what is meant by more effective therapies? What elements added to therapies would make them more effective and why?

Line 415: Spell out the author of citation for better readability.

2.2. Psycho-nutrition section: What is psycho-nutrition and when/how can it be used? Add more details and background about this concept.

2.3. Eating behavior section: Findings are presented in relation to designing therapies, emotions impacting eating behaviour and scale validation. Perhaps name this section differently to reflect on the content or revise the content to cover eating behaviour and what influences it.

Line 206: Add “as” after the word “such”.

Line 226: Use “such as” instead of “like as”.

2.4. Eating attitudes: bulimia and anorexia section: This section talks about the importance of early identification and treatment, and societal influences, but doesn’t discuss what are considered problematic eating attitudes and how these can lead to development of eating disorders if not detected early and what can be done to detect the signs.

Line 265: What does DSM-IV stand for? Can you spell this out or provide a definition to clarify the purpose and meaning?

Line 285: What exactly are the multiple assessment methods? Can you provide details?

2.6. Emotional food craving section: Description of other studies are provided but there are no links to their research findings beyond mentioning of the studies.

Line 334: This is a very specific example from a narrow population. How are these results relevant to your study as participants seem to be ordinary people?

Line 345: Example with a focus on population being pregnant women. How are these results relevant to your research?

2.7. Regulation of eating behaviour section: Similar to previous section, this one also provides acknowledgement of other studies without going into detail about the findings and how these would relate to the current study.

Line 370: Example discusses molecular processes and mentions providing insights into treating eating disorders but there’s no mention of what the insights are or what molecular processes refer to in this context.

Line 383: Media’s effect on women’s body issues is mentioned but again no deeper level explanation is provided. How do these examples connect to your audience? I would critically review the evidence used in literature review and reflect on what is relevant to the current study.

Methods: The paper attempts to describe the method used, but there's a lack of explanation on the constructs and/or variables used to determine dieting practices, attitudes, food cravings etc. The paper mentions that Google Form questionnaire was used to collect data but it’s unclear how the participants were recruited and if data collection was fully online. Additionally, there are no references to the questionnaire, questions are described very vaguely, and there’s no information on how the survey and questions were developed other than the mention of using some previously validated questions that haven’t been outlined.

Line 401: This paragraph would work better in the introduction section to set the scene for the study.

Line 406: The selection of the concepts needs to be explained in the introduction section.

Lines 415-419: This can be moved into the introduction section.

Line 424: Is this for Part 1 attitudes or Part 2 behavior? What are the standardised items? Can you create a table with all the items in it to provide more details and clarity?

Line 433: Spell out the authors of citations 72 and 58 for better readability.

Line 439: What are the previous studies? Citations are missing.

Line 440: What are the items? What scales were used? What aspects of food cravings were used?

Line 449: How were participants recruited? Online invitation, face to face etc.? Was ethics approval provided for this study?

Line 452: 80/20 split in gender is more than slight imbalance. This needs to be reflected in results, discussion and limitations.

Line 461: Why is height 1.52cm included? This is clearly incorrect and shouldn’t be included in any analyses as it skews the statistics.

Line 469: What are the key psycho-nutritional variables?

Results: The results are not presented clearly, partly due to a lack of description of the constructs and the questionnaire in the Methods section. The authors have included incomplete and incorrect data (e.g., person’s height being 1.52cm), which impacts the quality and correctness of the results.

Line 492: The first sentence doesn’t make sense. What does the statistical information for the subscales follow? Can you rewrite this?

Line 497: “values ranging from 13 to 78” what do these values mean? Low or high, good or bad. Can you explain the range? Same with the scales that follow this.

Line 508: Table 2, what is the meaning of these values? The same information is written above. I suggest you keep only one and clearly explain what the numbers mean as at the moment the scores on scales don’t provide explanation of what is going on or what they measure.

Line 509: In this section, you use different names for some variables e.g., emotional food wanting and emotional food craving. Use only one for consistency.

You explain the correlations, but can you also explain the ‘so what?’ e.g., what does it mean when you say that “higher dieting scores are linked to lower scores in these subscales”?

Line 539: Network analysis section: Can you explain what the numbers mean? Is there a certain threshold that’s required for betweenness and closeness, strength and expected influence?

Line 574: What do the line, colors and numbers in the figure mean? Please provide an explanation on how these relationships work.

Line 578: Sequential mediation analysis. Why were these pitted against each other? What does it mean when a relationship is significant?

Lines 611-614: This is well explained. Can you use similar ways to explain these results in other sections too?

Lines 638-640: What does this mean? What are the variables measuring?

Discussion: The paper discusses the interplay between food attitudes and the management of eating behavior from a psycho-nutritional perspective but struggles to build new ideas. It seems like the paper merely confirms the findings that have already been identified by previous research, and therefore it’s challenging to use the research in practice. There is no mention of theory use in the paper, limiting bridging the gap between theory and practice, influencing public policy or contributing to knowledge. The paper discusses the importance of behavioral interventions, but it is not clear how this paper influences attitudes or affects the quality of life. Implications are alluded to, but there is not a convincing argument.

The section mainly repeats what is communicated in introduction and literature review with findings from this study. Significant information relating to the research aims, background and data collection methods is missing that impacts the depth of the analysis.

Lines 644, 648, 657, 659, 661, 667, 670, 672, 679, 682, 685, 691, 693, 695, 698: Spell out the author of citation for better readability.

Conclusions: lines 717-719: There’s no need to report numbers here as these should be reported under results.

Lines 721-733: This would work better under discussion.

Lines 734-739: This would work better as a subsection after discussion and needs more details.

Overarching observations:

The study mentions no theory use and it’s unclear if the research was underpinned by any theory. The significance of the study remains unclear beyond confirming the results of previously conducted studies.

The study has potential and with thorough revisions highlighting the significance and providing a clear background it can add to the existing literature however, in its current form it’s not suitable for publication.

Reviewer 2 Report

Comments and Suggestions for Authors

The Intro discusses the findings from questionnaires/scales. Questionnaires or scales are tools which operationalize theoretical constructs. The authors need to convert their discussion to a discussion of the relations among theoretical variables.

There is an extensive literature on the relationships of psychological variables to dietary intake. The authors cite a fair amount of research (and there is a lot more) so why do the authors keep saying that the field needs research in this area?  

There is a literature on habit which should be cited and integrated into the discussion. See:Wendy Wood1, and Dennis RüngerPsychology of Habit, Annual Review of Psychology, Vol. 67:289-314 .

Simply analyzing data will not reveal truth. The authors need to propose a conceptual model of the variables and relationships in which they are most interested, supported by the theoretical and empirical literatures, and test for the proposed relationships in their data.

The authors need to provide a theoretical rationale for each of the scales employed, and cite validity coefficients in comparable samples to reassure the reader that the scales measure what is expected/desired.

The gender imbalance in the sample was not “slight”.

Table 1 should include all the “descriptive”/demographic variables and the dependent variables. 

Since most readers will not know the economics in Romania, can the authors also provide frequencies by categories of low, medium and high income or some other related categorization?

Did the authors calculate BMI or BMIz? Shouldn't they since BMI is related to mental health?

The variables in table 3 are very highly inter-related. The authors should conduct a factor or principal components analysis to assess the extent to which factors account for the variability in these scales. – It is possible that some of these values can be accounted for by confounding correlations with other variables, e.g. obesity status?

It appears to me that the authors collected a large number of variables, threw them into an analysis that could accommodate so many variables and churned out the numbers. This approach runs the risk of type one statistical error, detecting significance when the analyses violate the underlying assumptions. A more theory guided analysis with tests of specified relationships would have been preferred.